# Mammography screening: Eliciting the voices of informed citizens

**Manja D. Jensen**[1,2]*, **Kasper M. Hansen**[3], **Volkert Siersma**[1], **John Brodersen**[1,2]

**1** Department of Public Health, The Research Unit for General Practice and Section of General Practice, University of Copenhagen, Copenhagen, Denmark, **2** Primary Health Care Research Unit, Region Zealand, Roskilde, Denmark, **3** Department of Political Science, University of Copenhagen, Copenhagen, Denmark

* madj@sund.ku.dk

**Data Availability Statement:** Code and anonymised dataset files are available from the Harvard Dataverse: https://dataverse.harvard.edu/dataverse/MKVJ2.

## Abstract

### Background

Many medical organisations recommend continuing with existing mammography screening programmes but some recommend stopping or de-intensifying them. In Denmark women aged 50–69 are offered biennial mammograms free-of-charge.

### Objectives

The aim of this study was to determine whether or not an informed public would recommend continuation of the Danish mammography screening programme, and to determine whether this recommendation was in line with what participants considered to be acceptable levels of mortality reduction and overdiagnosis.

### Methods

A Deliberative Poll on mammography screening was held online in Denmark and 89 citizens participated. They were representative of the general population on sociodemographic parameters, attitudes towards and knowledge of mammography screening. Participants studied a video about the programme and took part in an online citizens' assembly where they deliberated with peers and experts in the field. All participants answered a survey at four time points: at recruitment; after video information; after deliberation, and a month after the assembly.

### Results

Questionnaire data revealed that many participants were influenced by the deliberative polling process as 36%, changed their recommendation afterwards. At recruitment, 72% of participants strongly supported the continuation of mammography screening. This proportion was lower after the presentation of video information (55%), after deliberation (65%), and a month after the assembly (58%). Interestingly, these changes in recommendation were not correlated to changes in knowledge. The proportion of participants who recommended continuation following what they stated were acceptable rates of mortality reduction and overdiagnosis rose from 21% at recruitment to 40% after information and deliberation. Most

**Funding:** The project was funded by: Region Zealand, Den forskningsfremmende pulje (JB, www.regionsjaelland.dk), Region Zealand, PhD grant (MDJ, www.regionsjaelland.dk), Helsefonden (MDJ, 17-B-0238, www.helsefonden.dk); Fonden for almen praksis (MDJ, A1525, www.laeger.dk/fonden-for-almen-praksis); Poul og Agnes Friis fond (MDJ, 81008-001); Lilly og Herbert Hansen fond (MDJ, 051), and A.P Møllers lægefond (MDJ, 18-L-0021, https://www.apmollerfonde.dk/). The funders had no role in the study design, data collection and analysis, decision to publish, or preparation of the manuscript.

**Competing interests:** The authors have declared that no competing interests exist.

participants (60%), therefore, made a recommendation that was not in line with levels of mortality reduction and overdiagnosis that they felt were acceptable.

## Conclusion

After video information and deliberation participants were less supportive of the mammography screening programme compared to their immediate recommendation at the beginning of the process.

---

"*It is better to debate a question without settling it*

*than to settle a question without debating it*"

*(Joseph Joubert, 1850)* [1]

## Introduction

Mammography screening is much debated. Inviting healthy women to participate with the intention of reducing mortality from breast cancer can cause severe harm in terms of overdiagnosis, and frequent harm in terms of false positive results. The main contention in medical discussions is whether the intended benefits of the screening programme outweigh the unintended harms.

Currently all women in Denmark aged 50–69 have the right to biennial, free-of-charge mammography screening. Many countries have implemented such programmes which rely on the ability to meet national criteria [2,3] based on the pioneering work of Wilson and Jungner [4,5]. This evaluation concerns whether benefits exceed harms, and it also addresses the quality of the screening test, whether the disease being screened for constitutes a considerable health problem in society, and other aspects such as economic considerations. It is appealing to argue that evidence can provide the answer as to whether or not a screening programme should be implemented, but such judgements involve values as well as evidence. Experts evaluate the evidence differently and make different value judgements [6]. This is clear looking at guidelines on mammography screening published by major medical organisations. They vary regarding target age range, screening interval, and whether breast screening should be offered at all [7]. Many medical organisations recommend mammography screening, including the US Preventive Service Task Force (USPSTF) [8] and the World Health Organization (WHO) [9], among others, and both the EU Commission and the German Institute for Quality and Efficiency in Health Care (IQWIG) recommend extending the programme [10,11], However, the Swiss Medical Association, The Cancer Expert Working Group of Hong Kong, and an independent expert group in France recommend stopping or de-intensifying mammography screening [12–14].

Public engagement is increasingly recognised as an integral part of healthcare [15,16]. When the public is engaged in healthcare matters they are often constructed in one of three ways: as advocates (constituting experts and interest groups); as consumers (the affected public), or as citizens (the "pure" public) [17]. In a Danish context, patients and consumers are continuously involved in healthcare through satisfaction surveys and interviews to give

insights about their views on different interventions [18]. In terms of political decisions on healthcare, citizens' opinions are largely represented by patient associations under the umbrella organisation "Danish patients" [19]. Citizens also participate in the development of healthcare policy in their local municipalities in the capacity of being citizens through so-called deliberative approaches [20]. If, why, when and how citizens should be involved in policy matters related to screening can be debated. Public engagement in decision-making or guideline development about screening could supplement the different preferences of experts regarding the balance between benefits and harms [21]. Directly asking the public about their opinions on a screening programme, however, might result in responses based on non-attitudes and misconceptions. Information given to the public has been predominantly one-sided in favour of screening [20–25]. Studies have demonstrated limited public awareness and understanding of the serious harms of overdiagnosis [26–29]. Public belief in the benefits of screening are, in general, exaggerated, and their belief in the harms are understated [30,31].

Deliberative methods can be employed [17], to ensure consideration of both citizens' values and scientific evidence when eliciting public recommendations, including Citizens' Juries, Deliberative Polls, and consensus conferences, allowing for thorough dissemination of information, understanding and discussion of complex dilemmas among citizens [21,32]. These methods are well-established in fields outside healthcare and have also been successfully applied to different issues within healthcare [32]. Regarding mammography screening, Citizens' Juries have been the preferred deliberative approach [33–36]. A Citizens' Jury typically engages around 15 citizens. Participants are provided with information by experts in the field and deliberate over several days assisted by moderators. Citizens' Juries can take different approaches. Participants can vote between a set of options or they can formulate recommendations in response to broad, open-ended questions, or both [34,37,38]. Studies on Citizens' Juries concerning mammography screening have used qualitative methods to provide valuable information about the reasons for different recommendations and votes [33–36]. However, as the number of participants is often relatively low, limited representativeness of the wider population does not allow for generalisation of recommendations and the range of arguments is often limited. A Deliberative Poll, on the other hand, gives an aggregated opinion of what the public would think if they had all been given the opportunity to engage in a process of deliberation and information [39]. A Deliberative Poll is characterised by bringing a representative group of people together. Participants deliberate with each other and with experts for one or several days and their discussions are based on the presentation of balanced information. Using a survey, their opinions are polled before, during, and after the process [40]. This makes it possible to link knowledge, opinions and recommendations and to explore possible changes as a result of information and deliberation.

Deliberative methods aim to elicit informed and considered opinions and rely on normative beliefs about the dynamics of a group: sharing of information and thoughts in a group will facilitate new ideas clarify questions and increase knowledge. In addition, everyone in a group will be encouraged to reflect on and give reasons for their views, and respond to the arguments of others. The more information and the more arguments participants are presented with during the process, the less likely it is that they will encounter new information and arguments in the future which could alter their opinions on the subject. When participants must reflect on (new) information and arguments they might uncover inconsistencies in their previous arguments or opinions. Inconsistent arguments are hard for others to follow, therefore, group deliberation is believed to force participants to think things through [41].

Our previous study focused on whether or not our experimental setup—the Deliberative Poll—approached the purpose or potential of this type of citizens' involvement in decision-making regarding continuing the Danish mammography screening programme. We analysed

how information and deliberation affected the quality of decisions among participants using the following outcomes: Knowledge; Ability to form opinions; Opinion stability, and Opinion consistency [42].

Knowledge was conceptualised as correct answers to 13 items addressing conceptual and numeric knowledge. Participants' ability to form opinions was conceptualised as their ability to answer opinion questions with anything other than 'don't know'. Stability was operationalised as correlations *within* opinion items at *different* time points; whereas consistency was operationalised as correlations *between* opinion items *within* each timepoint.

We found that participants' knowledge about mammography screening increased markedly because of the Deliberative Polling process; at recruitment, only 1% of participants were able to answer two-thirds of the knowledge questions correctly. After video information, the proportion was 56% [42]. Knowledge increased on most items (11 out of 13) (S1 Table) [42]. We also found that participants' ability to form opinions about aspects related to mammography screening (opinion items) increased as a result of the process [42]. In addition, opinion stability and opinion consistency increased [42].

The present study focused on how information and deliberation affected citizens 'recommendation about continuing the Danish mammography screening programme. The primary aim was to determine an informed public's recommendations about the continuation of the Danish programme, and also to determine whether an informed recommendation was in line with what participants considered to be acceptable levels of mortality reduction and of overdiagnosis in a mammography screening programme (referred to as their *preferences* regarding mortality reduction and overdiagnosis). The secondary aim was to examine how the process of information and deliberation affected opinions related to mammography screening.

Assuming that people's recommendations about screening on a population level will be based on and modified by knowledge about the programme, and bearing in mind that laypeople often overestimate the benefits and underestimate the harms of screening, we hypothesised: 1) That there would be a decrease in support for the continuation of mammography screening after information and deliberation about the benefits and harms of the programme compared with levels of support before the intervention; 2) That a decrease in support would be correlated to an increase in knowledge. Our third hypothesis is based on beliefs about how group dynamics can affect inconsistent arguments and opinions. Taking this into account and considering that the Deliberative Poll process includes the presentation of information concerning the benefits and harms of the programme, we hypothesised 3) That participants would make recommendations more in line with their stated preferences regarding mortality reduction and overdiagnosis after information and deliberation.

## Method

We adopted the Deliberative Poll as our deliberative method [39,43]. The available methods have previously been described in detail elsewhere [42]. In short, a Deliberative Poll involves a group of citizens who are representative of the general population in terms of demographics, opinions and sample size. Participants are provided with information material to form the basis for deliberation with other citizens, as well as with experts in the field. Before, during and after the deliberative process, participants' opinions are assessed by a survey. Changes in opinion between the first survey at recruitment and subsequent surveys after information and deliberation represent the changes the general public would experience if everyone had the same opportunity to be informed and to deliberate about the issue at stake.

## Recruitment of participants

Kantar Gallup, an independent polling company, ran the recruitment process. Participants were recruited through stratified random sampling. Kantar Gallup used their large online panel of 50.000 randomly recruited citizens. They emailed 1977 men and women from this panel with a link to an online questionnaire. The citizens were asked to take part in a "a citizens' assembly concerning mammography screening as part of a research project". Participants were excluded if they did not respond, if they declined to be further contacted, if they did not want to participate in the citizens' assembly, if they had no camera/microphone on their computer or tablet (necessary for the online citizens' assembly), or if they exceeded their population quota (e.g. percentage of woman, specific age-groups etc. corresponding to the general Danish population.) Participants were paid 500 DKK (67 €) in compensation for their time. Following recruitment, a mini-public of 89 participants attended a citizens' assembly and Deliberative Poll on mammography screening. See Fig 1 for flow diagram. Participants were representative of the Danish population aged 18–70, based on the sociodemographic parameters of gender, age, educational level, marital status, and residence; and on the additional

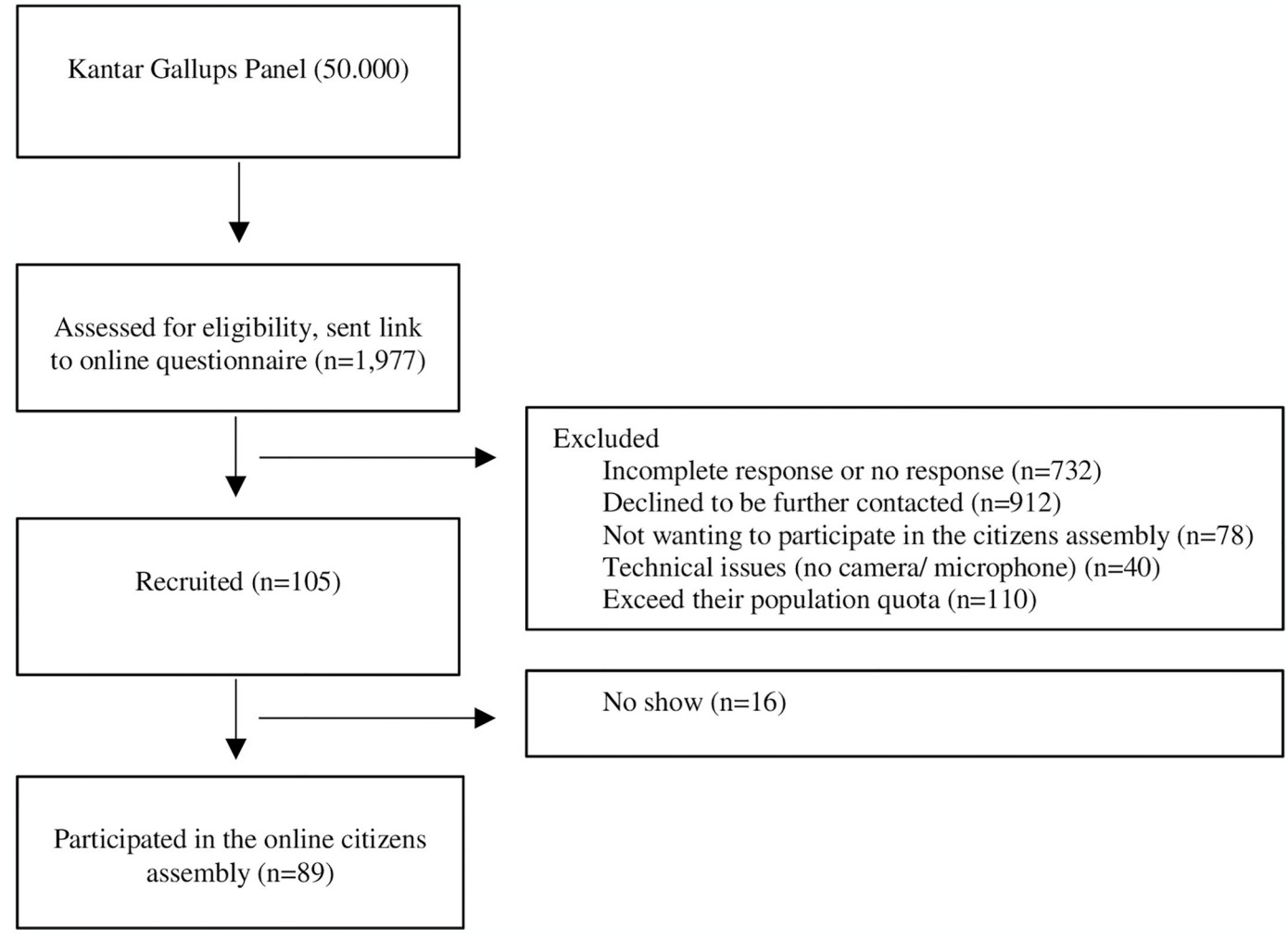

**Fig 1. Flow diagram for recruitment.**

parameters: worry, family history of breast cancer, education within healthcare, knowledge, and opinions about mammography screening (Table 2).

## The deliberative polling process

On Sunday the 20th of September 2020 a citizens' assembly and Deliberative Poll about mammography screening was held online in Denmark. Prior to the assembly, participants were encouraged to study a twenty-minute-long information video about mammography screening, which was produced for this Deliberative Poll. In addition to the video, participants were given a fact sheet summarising the video information. An English version of the fact sheet has previously been published [42].

At the online citizen's assembly, participants were greeted by the former chief of the Danish Council on Ethics who chaired the assembly. The day began by showing the video and the participants then deliberated in small group sessions (eight participants in each) led by moderators, primarily schoolteachers. Subsequently, participants returned to plenum and had the opportunity to ask questions of three external experts in the field: an anthropologist, a medical doctor, and a health economist. The first session of group discussion and plenum discussion concerned understanding the video and the concepts presented. A second session of group deliberation and plenum discussion held in the afternoon concerned balancing benefits and harms. In both group sessions the aim was to formulate questions to ask the experts in the following plenum discussion with the intention of having the participants rather than the experts setting the agenda for deliberation. To reduce the likelihood of undesirable group dynamics several precautionary measures were taken. The group moderators were instructed to take a neutral position, to ask participants not to introduce themselves in terms of their job, and to encourage participants to keep an open mind and not state their opinions and recommendation about the programme at the outset. This was done to minimise social positioning and psychological entrapment. There was no a priori aim of reaching a consensus, which could have increased the risk of conformity and groupthink.

## Ethics

According to the Committees on Health Research Ethics for the Capital Region of Denmark (De Videnskabsetiske Komiteer for Region Hovedstaden) the project does not constitute a health research project, but is considered a questionnaire-based study as defined by the "Danish Act on Research Ethics Review of Health Research Projects", Section 2. Thus, this project is not subject to notification from the Committees (Journal-no.: 21031705).

The Office of Research and Innovation at the University of Copenhagen has approved the study, reference 514-0385/19-3000 [44]. Electronic informed consent was obtained in the first questionnaire where participants were provided with information about the project. Consent was given by answering "Yes, I accept" and also by providing preferred contact information in the online questionnaire.

## Material

### Video information

Prior to the Citizens' Assembly and Deliberative Poll, a video about mammography screening was created. Communicating nuanced and understandable information about mammography screening is difficult. There is much debate about which facts to present, what constitutes the best available evidence and in which format to present risk information [45–49]. In a previous study, we described in detail our information material including how we presented numbers

as natural frequencies and used icon arrays formatted consistently with a shared common reference class [42].

In short, the video provided evidence-based information about the benefits and harms of mammography screening and began by correcting some common misunderstandings, such as: "all cancers will progress and become symptomatic" and "it always makes a difference to find and treat cancer earlier". Women in Denmark could experience mammography screening at least 10 times from the age of 50 to 69 years, and so the video talked about the benefits and harms of the programme over twenty years. The video presented the effects of inviting women, e.g. adjusted for an 83% participation rate. Estimates of mortality reduction and over-diagnosis were derived from a Cochrane review [50], which was chosen as the basis for these calculations as it is transparent about bias assessment. The number of false positive and false negative screening results was derived from the actual programme based on The Danish Quality Database of Mammography Screening (Annual Report 2019) [51]. Mortality and incidence data were drawn from Statistics Denmark [52] and costs were drawn from an economic evaluation of the Danish programme [53]. The video can be accessed at: http://bit.ly/mammografi-screening..

## Questionnaire, data collection and trial outcomes

At four time points: at recruitment ($T_1$), after video information ($T_2$), after deliberation ($T_3$), and one month after the citizens' assembly ($T_4$), participants' knowledge, opinions, and recommendations were assessed using an online questionnaire. Knowledge was assessed by 13 knowledge items concerning conceptual and numerical aspects of mammography screening. The items covered the benefits and harms of the programme introduced in the video and fact sheet. This information functioned as "ground truth". A multiple-choice design was used to assess knowledge, where participants choose between different answers and "don't know" [42]. See Table 1 for knowledge items. Opinion items covered some cognitive, affective and social aspects of screening that could relate to participants' recommendations about the future of the mammography screening programme in Denmark. The opinion and recommendation items consisted of statements. The participants indicated their position in respect of the statements on a 5-point Likert scale [42]. See Table 3 for the opinion items and Fig 2 for the wording of the recommendation item (whether we should continue mammography screening in Denmark). Two items (multiple-choice design) assessed participants' view on what they considered to be acceptable levels of mortality reduction and of overdiagnosis in a mammography screening programme. See Fig 3 for the wording of the items. The questionnaire was tested on three laypeople and adjusted in "think-aloud" tests.

## Statistical analysis

An a priori power analysis determined a sample size of a minimum of 100 participants to achieve statistically significant results, with an over 10 percentage point change in key items. The sample size was calculated bearing in mind that we were testing the same people before, during, and after the intervention with a paired test and we assumed a correlation of 0.6 between responses. Robust evidence [26,27,31] has revealed that laypeople overestimate the benefits and are unaware of or underestimate the harms of screening, thereby basing their support of screening programmes on biased assumptions. Therefore, we hypothesised that fewer people would favour continuing mammography screening after the intervention compared with before.

Our previous study documented changes in knowledge (typically knowledge gain) and changes in opinions as a result of the deliberative polling process [42]. In the present study, the

**Table 1. Knowledge items.**

| **Conceptual items** |
| --- |
| *The following items are about breast cancer and mammography screening in women age 50 to 69 years. Do you find the following statements to be true or false? There are many difficult questions about mammography screening. Remember that you can also answer don't know.* |

Screening is for women without symptoms
 True
 False
 Don't know

Not all breast cancers cause illness
 True
 False
 Don't know

Screening reduces breast cancer deaths
 True
 False
 Don't know

Screening increases breast cancer diagnoses
 True
 False
 Don't know

Screening leads to some women getting unnecessary treatment
 True
 False
 Don't know

Screening will not find every breast cancer
 True
 False
 Don't know

What does a false positive result mean?
 The mammogram looks abnormal but further test reveal that there is no cancer in the breast
 The mammogram shows breast cancer but further test reveal that there is no cancer in the breast
 The mammogram looks normal but after the examination the woman finds a lump and is diagnosed with breast cancer.
 Don't know

Mammography screening results in some women having more years of life with a breast cancer diagnosis. What does this mean?
 Screening results in more breast cancers being diagnosed and treated. It is in this context we talk about "more years of life with a breast cancer diagnosis". It is therefore considered a benefit of the programme.
 Screening alters the time of diagnosis; the cancer is diagnosed at an earlier point in time. In some cases, this will mean that a woman will live with a breast cancer diagnosis for more years without an effect on her survival, whether or not she will die of breast cancer. Because living with a breast cancer diagnosis can have negative effects on one's life "living longer as a patient" is considered a harm.
 Don't know

"Benefit evaluation—reduced mortality"
The beneficial effect of screening can be assessed looking at:
 For how long women live with a breast cancer diagnosis
 Mortality reduction
 The number of women getting a breast cancer diagnosis
 Don't know

| **Numerical items**[*] |
| --- |
| *In relation to the next questions we will ask you to imagine 1000 women from age 50 to 69.* |

"Breast cancer mortality without mammography screening"
Imagine that these 1000 women are not screened. How many will die from breast cancer?
 Write a number
 Don't know

*(Continued)*

**Table 1.** (Continued)

| |
| --- |
| "Breast cancer mortality with mammography screening" |
| Imagine that these 1000 women are screened every second year in a 20 years period. How many will die from breast cancer? |
| Write a number |
| Don't know |
| "Overdiagnosis" |
| Imagine that these 1000 women are screened every second year in a 20 years period. How many will be overdiagnosed with breast cancer? |
| Write a number |
| Don't know |
| "False positives" |
| If the women are screened every second year in a 20 years period some will experience false alarms. How many? |
| Write a number |
| Don't know |

*Numbers accepted as correct: Breast cancer mortality without mammography screening: 8-14/1000 women in a 20-year period, Breast cancer mortality with mammography screening: 8-14/1000 women in a 20-year period offered screening every second year, Overdiagnosis: 8-14/1000 women in a 20-year period offered screening every second year. False positive: 100-200/1000 women in a 20-year period offered screening every second year.

marginal associations between the change in recommendation due to the deliberative polling process, i.e. the difference in the recommendation item score from T1 (recruitment) to T3 (after information and deliberation), and concurrent changes in the knowledge index, individual knowledge item scores and opinion item scores, were assessed with Pearson correlation coefficients.

Further, to investigate which aspects of change in knowledge and opinion, respectively, are most important to a change in recommendation, the relative importance of the changes in the collected knowledge and opinion items, resp., was calculated in dominance analyses [54]. Such analyses divide the coefficient of determination ($R^2$) from a complete multivariable linear regression model that includes all changes in items as main effects into the parts attributable to each of the changes in items. This is done by, for each change in item, averaging the increase in $R^2$ obtained by adding the change in item to the model, over all models that can be constructed by including subsets of the other changes in items as main effects in the model. This results for each change in item in a percentage of the $R^2$ attributable to this change in item; changes in item that account for a high percentage rank higher in importance than changes that account for a low percentage.

To determine which opinion carried most weight for the recommendation at each of the stages in the Deliberative Poll, the relative importance of the opinion items to the recommendation at each of the four inquiry time points was assessed in dominance analyses similar to the above described, and visualized with pie charts.

A paired chi-squared test was used to test the change in acceptance of high levels of overdiagnosis as well as acceptance of a mortality reduction of one between $T_1$ and $T_3$.

## Results

The study succeeded in recruiting a sample that was statistically representative of the adult Danish population on sociodemographic parameters as well as on attitudes to and knowledge of mammography screening (Table 2).

Participants' recommendations changed between the four inquiry timepoints: at recruitment ($T_1$) 72% strongly agreed to continue mammography screening, while 55% strongly

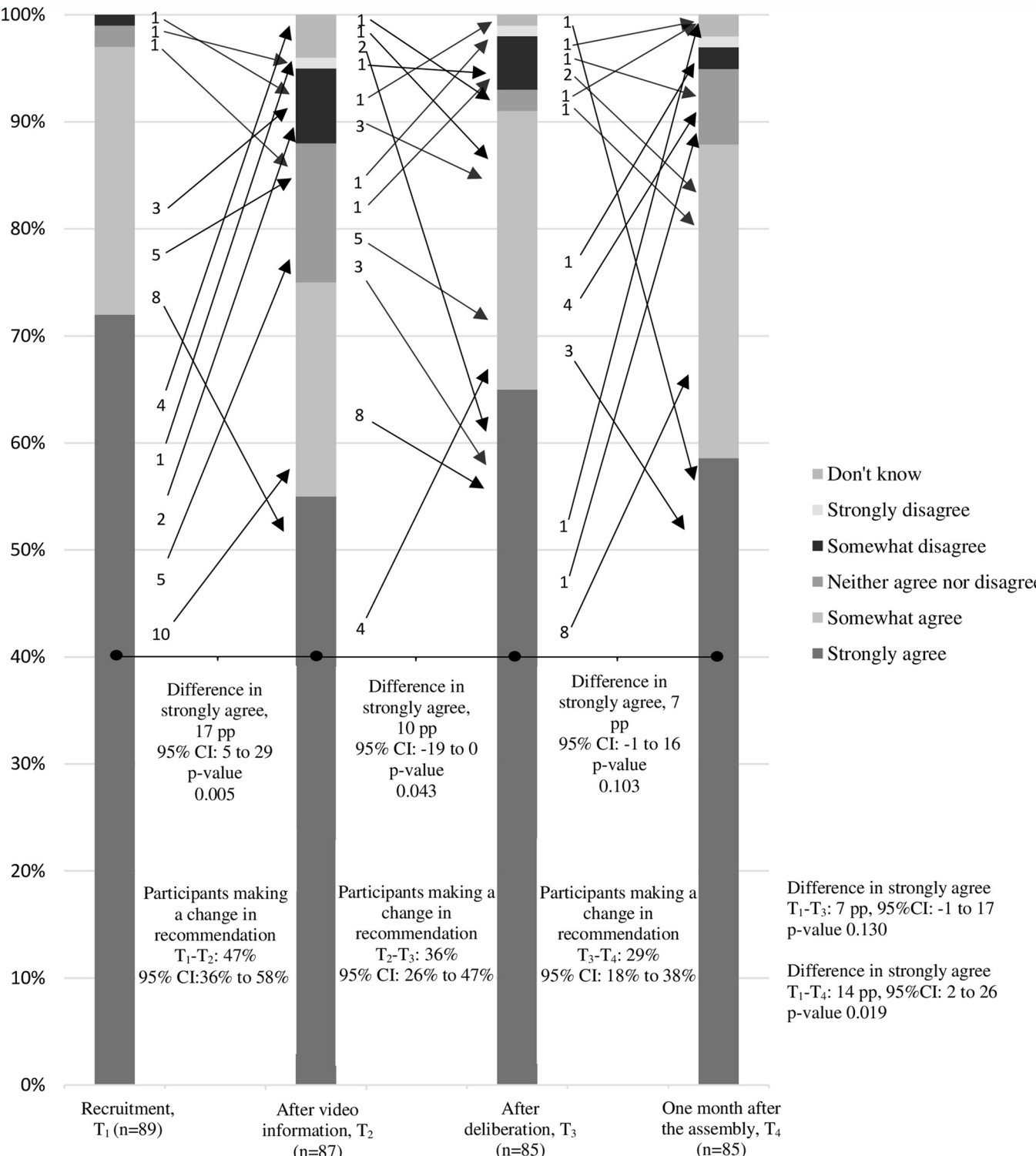

**Fig 2. Recommendation "Continue mammography screening".** Note: The bar chart display participants' responses to the question "To what extent do you agree that we should continue mammography screening in Denmark?". The arrows with numbers indicate specific changes in recommendation between time points.

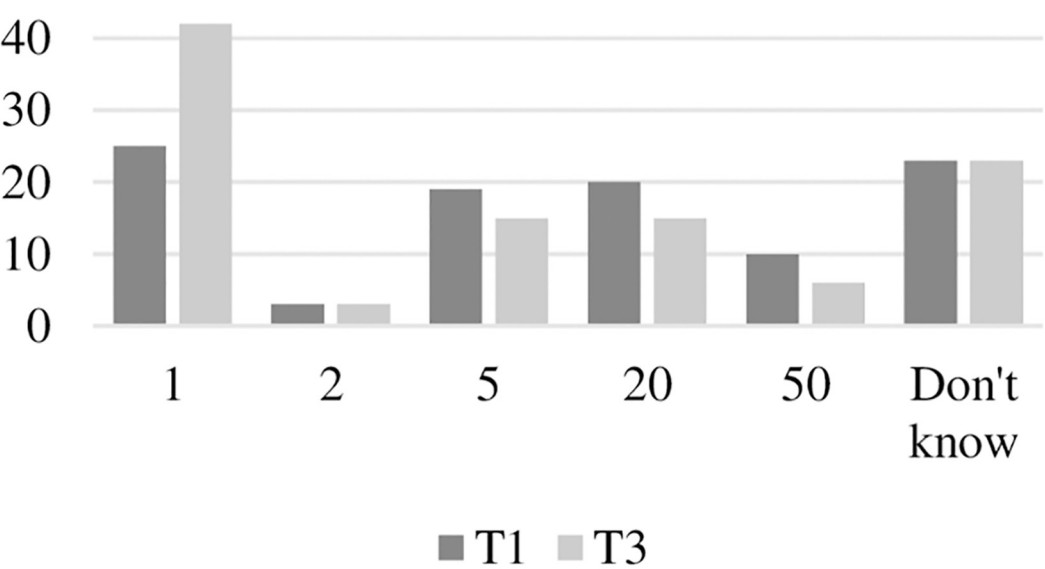

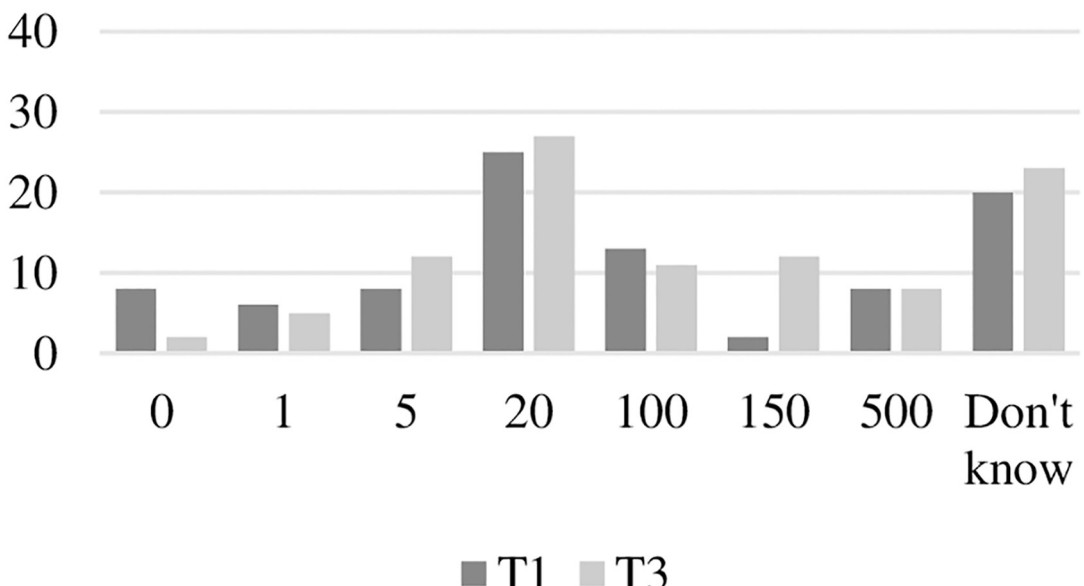

**Fig 3. Acceptable rates of mortality reduction and overdiagnosis.** Note: The bar chart shows the percentage of participants accepting different thresholds for mortality reduction (multiple-choice). $T_1$ = inquiry time point 1 (recruitment) n = 89, $T_3$ = inquiry time point 3 (after deliberation) n = 85. Change in acceptance of a mortality reduction of 1 compared to any other answer (2, 5, 20, 50, Don't know), between T1 and T3 was tested with a paired t-test; 17.6 percentage point (95%CI: 6.1 to 29.2), p-value 0.003. Note: The bar chart shows the percentage of participants accepting different levels of overdiagnosis (multiple-choice). $T_1$ = inquiry time point 1 (recruitment) n = 89, $T_3$ = inquiry time point 3 (after deliberation) n = 85. Change in acceptance of high levels of overdiagnosis (100,150,500) compared to low levels (0,1,5,20), between T1 and T3, was tested with a paired t-test; 10.4 percentage point (95%CI: -3.1 to 23.9), p-value 0.1364.

**Table 2. Participants' characteristics (%).**

| Characteristics | | Participants (n = 89) % | Eligible Danish population, age 18–70 years (n = 3.885.797) % | Strongly agreeing to continue the programme at $T_1$ (n = 64) % |
|---|---|---|---|---|
| Gender | Men | 52 | 50 | 65 |
| | Women | 48 | 50 | 79 |
| Age group | 18–30 years old | 20 | 25 | 61 |
| | 31–40 years old | 19 | 18 | 59 |
| | 41–50 years old | 24 | 20 | 71 |
| | 51–60 years old | 20 | 20 | 83 |
| | 61–70 years old | 17 | 17 | 87 |
| Education | 7–13 years in school | 41 | 36 | 69 |
| | Vocational training | 28 | 29 | 76 |
| | Short further education | 8 | 5 | 71 |
| | Middle further education | 13 | 18 | 75 |
| | Long further education | 10 | 12 | 67 |
| Residence | Capital Region | 42 | 32 | 71 |
| | Region Zealand | 14 | 14 | 67 |
| | Region of Southern Denmark | 18 | 20 | 81 |
| | Central Denmark Region | 19 | 23 | 71 |
| | North Denmark Region | 7 | 10 | 67 |
| Marital status | Never married | 50 | 42 | 67 |
| | Married or civil partnership | 32 | 44 | 75 |
| | Divorced or separated | 14 | 12 | 75 |
| | Widowed | 4 | 2 | 100 |
| Characteristics Data available from a pre-intervention measure | | Participants (n = 89) | Pre-recruitment (n = 1,112–1,290) | |
| Worry about breast cancer—never | | 31 | 33 | 56 |
| Working/educated within the healthcare sector | | 15 | 15 | 85 |
| History of breast cancer in the family | | 37 | 37 | 70 |
| Conceptual knowledge | | | | |
| Diagnosis, correct answers (Screening increases breast cancer diagnosis) | | 68 | 62 | 75 |
| Overtreatment, correct answers | | 16 | 14 | 64 |

Note: The table shows the sociodemographic characteristics in the eligible Danish population and in participants of the Deliberative Poll. Data from the eligible Danish population of 18-70-year-olds available from Statistics Denmark. No statistically significant difference was found between the sample of participants and the eligible population using Chi$^2$-test in any of the above factors.

agreed after watching video information ($T_2$). After deliberation ($T_3$) 65% strongly agreed to continue mammography screening and one month later ($T_4$) 58% strongly agreed. The greatest change in recommendation happened at the beginning of the Deliberative Poll ($T_1$-$T_2$) where 47% changed their recommendation. After the end of the poll, between $T_3$-$T_4$, 29% changed their recommendation (Fig 2).

*"It is discussed among researchers if it is worthwhile conducting mammography screening in Danish society. Some think the benefits exceed the harms. Others that the harms exceed the*

*benefits. Others again think the programme is not worth the expense. Others think it is. In the following questions, we will ask you to consider mammography screening as an offer to women in Denmark. To what extent do you agree that we should continue mammography screening in Denmark?"*

Learning (change in knowledge index) was not correlated to a change in recommendation, correlation 0.03 (95%CI -0.19–0.24) (Table 3). On the contrary, a change in opinion towards viewing the harms of screening carrying more weight than the benefits, was correlated to a change in recommendation towards being more sceptical. In addition, a change towards viewing mammography screening as having little effect on breast cancer mortality was also correlated to a change in recommendation towards being more sceptical. Changes in five opinion items were correlated to a change in recommendation towards being more positive about the programme: 1) being more in support of the programme; 2) being more in agreement with the health authority's recommendation; 3) to a larger degree viewing costs as reasonable; 4) to a larger degree viewing health authorities as being obliged to provide mammography screening; and 5) to a larger degree feeling that Denmark would regret abolishing mammography screening (Table 3).

A change in opinion about the balance between benefits and harms was of relatively little importance to change in recommendation (6%) (Table 3).

Participants with a high level of knowledge at recruitment were more inclined to change their recommendation towards a more sceptical position compared to participants with lower levels of knowledge at recruitment OR 4.83 (1.40;16.65) (S2 Table). There was no statistically significant difference in the likelihood of change in recommendation towards a more sceptical position between various sociodemographic subgroups (S2 Table).

The relative importance of different opinion items in relation to recommendations changed during the process (S1 Fig). Video information affected some of the relationships between opinions and recommendations while deliberation affected others. At recruitment ($T_1$) participants' opinion about whether Denmark would regret it if the mammography screening programme were abolished was best at predicting recommendation. After video information ($T_2$) deliberation ($T_3$) and one month after the assembly ($T_4$), participants' opinions about the balance between benefits and harms, their opinions about the costs of the programme, and their views on health authorities' obligation to provide mammography screening increased in importance compared with $T_1$. Participant worry had a relative importance of 8% at $T_1$ but contributed only 1% or 0.5% at the following three inquiry time points (S1 Fig).

During the process of information and deliberation participants changed their view on what they considered to be an acceptable mortality reduction from mammography screening. At recruitment ($T_1$) 25% accepted one avoided death per 1000 women invited to screening over 20 years. After information and deliberation ($T_3$) the percentage was 42% (Fig 3). There was a tendency towards acceptance of more overdiagnosis after information and deliberation ($T_3$) compared with at the time of recruitment ($T_1$) (Fig 3).

*Imagine 1,000 women from age 50 to 69 years old. They are screened every second year. How many of those women are to avoid death from breast cancer before you would find it acceptable for the programme to continue?"*

*"Imagine 1,000 women from age 50 to age 69 years old. They are screened every second year. How many overdiagnoses can you then accept if one woman avoids dying of breast cancer because of screening?"*

**Table 3. The correlations between *change* in recommendation and *change* in knowledge and opinions between $T_1$ and $T_3$.**

| Knowledge | | Correlation coefficient (95% CI) for *change* in knowledge index/item and *change* in recommendation | Relative importance, % of *change* in knowledge item and *change* in recommendation |
|---|---|---|---|
| Knowledge index | (all 13 items below, see note) | 0.03 (-0.19;0.24) | |
| Separate items | | | |
| (short-form) | Screening is for women without symptoms | -0.01 (-0.23;0.20) | 8 |
| | Not all breast cancers cause illness | -0.02 (-0.23;0.20) | 1 |
| | Screening reduces breast cancer deaths | 0.08 (-0.14;0.29) | 10 |
| | Screening increases breast cancer diagnoses | 0.14 (-0.07;0.34) | 6 |
| | Screening leads to some women getting unnecessary treatment | 0.08 (-0.13;0.29) | 3 |
| | The meaning of false positive results | 0.03 (-0.19;0.24) | 6 |
| | Screening will not find every breast cancer | -0.07 (-0.28;0.15) | 5 |
| | Screening may result in prolonged life as a patient | 0.17 (-0.05;0.37) | 20 |
| | Benefit evaluation—reduced mortality | -0.11 (-0.32;0.10) | 9 |
| | Breast cancer mortality without mammography screening | -0.02 (-0.23;0.19) | 1 |
| | Breast cancer mortality with mammography screening | 0.04 (-0.18;0.25) | 13 |
| | Overdiagnosis | 0.00 (-0.21;0.21) | 8 |
| | False positives | -0.15 (-0.35;0.07) | 10 |

| Opinion item | | Correlation coefficient (95% CI) For *change* in opinion item and *change* in recommendation | Relative importance, % of *change* in opinion item and *change* in recommendation |
|---|---|---|---|
| Balance | see below for full question wording | -0.38 (-0.55;-0.18)* | 6 |
| Worry | I'm worried that I or someone in my family will die of breast cancer | 0.10 (-0.12;0.30) | 2 |
| Support | Some people are in support of mammography screening where others are not. Are you in… | 0.62 (0.47;0.74)* | 34 |
| Authorities | To what extent do you agree with the health authorities' recommendations about mammography screening | 0.31 (0.11;0.49)* | 4 |
| Politics | To what extent do you agree with political decisions about mammography screening in general | 0.09 (-0.12;0.30) | 1 |
| Effect | Mammography screening has little effect on breast cancer mortality | -0.37 (-0.54;-0.17)* | 7 |
| Costs | Economic costs of mammography screening are reasonable | 0.37 (0.17;0.54)* | 11 |
| Qualified | Citizens like myself are qualified to take a stand on whether women in Denmark should be invited for mammography screening | 0.19 (-0.03;0.38) | 1 |
| Mandatory 1 | It's mandatory to participate in mammography screening | 0.08 (-0.13;0.29) | 3 |
| Mandatory 2 | Health authorities are obliged to provide mammography screening to 50-69-year-old women in Denmark | 0.50 (0.27;0.61)* | 14 |
| Ethics | There are ethical dilemmas regarding screening mammography | -0.04 (-0.25;0.17) | 1 |
| Acquaintances | Most of the people important in my life think we should have mammography screening in Denmark | 0.17 (-0.04;0.37) | 1 |
| Regret | Denmark would regret abolishing mammography screening | 0.47 (0.28;0.62)* | 15 |

*(Continued)*

**Table 3.** (Continued)

| Seen | In general, I feel seen and heard in the healthcare system | 0.19 (-0.03;0.39) | 2 |
|------|------|------|------|

Note: The *change* in recommendation from $T_1$ (recruitment) to $T_3$ (after information and deliberation) was related to concurrent *change* in knowledge and opinions with Pearson correlation coefficients. Relative importance was calculated in a dominance analysis. The knowledge index combines all 13 questions giving 7.69 points for each correct answer. The index ranges from 0 to 100. See Table 1 for question wording related to the knowledge items. Question wording related to the item "Balance": Below are two statements. You can think of them as a discussion between two people, A and B. We will ask you to answer if you are most in agreement with A or most in agreement with B. Even if you do not totally agree with any of the positions, we will ask you to answer which position comes closest to your own viewpoint. The question is: How do you consider the balance between benefits and harms in mammography screening? A says: The harm carries most weight. Too many women are overdiagnosed and overtreated, and getting false alarms compared to the women prevented from dying of breast cancer. B says: The benefit carries most weight. The number of women prevented from dying of breast cancer counterbalances the number of women who are overdiagnosed and overtreated, and those getting false alarms. "Knowledge": $R^2$ = 46.9, "Opinion":$R^2$ = 58.0.

At recruitment ($T_1$), 21% of participants made a recommendation in accordance with what they stated were acceptable rates of mortality reduction and with what they stated were acceptable levels of overdiagnosis. After information and deliberation ($T_3$) this rose to 40% (S3 Table).

Accepting a high level of overdiagnosis was associated with a strong recommendation to continue the mammography screening programme, OR 5.23 (1.52;17.99). There was a tendency for increased acceptance of high levels of overdiagnosis and a strong recommendation in favour of mammography screening among participants with a low level of knowledge about overdiagnosis (S2 Fig).

## Discussion

### Change in recommendation

Many participants were affected by the Deliberative Polling process as 47%, 36%, and 29% respectively changed their recommendation between the four inquiry time points (Fig 2). Our findings are in line with our first hypothesis but not our second: that fewer participants would be supportive of continuation of mammography screening after information and deliberation (Fig 2); however, this change was not correlated to changes in knowledge (Table 3). In line with our third hypothesis, recommendations were more in line with what participants considered to be acceptable rates of mortality reduction and acceptable levels of overdiagnosis in a mammography screening programme after information and deliberation compared with before.

### Understanding change in recommendation

Different theories may help to explain the finding that change in recommendation was not correlated to change in knowledge.

We considered theories that deal with how individuals arrive at a recommendation based on cognitive processes (dual process theory and cognitive dissonance theory), as well as views about how opinions could relate to participants' social and societal context. If participants use mental shortcuts or gut feelings to varying degrees to ease the cognitive load of decision-making, instead of reflecting on knowledge and values [55], one could not expect a correlation between a change in knowledge and a change in recommendation. As mammography screening has run for more than ten years in Denmark, with high participation rates and the backing of high profile interest groups, a gut feeling in favour of screening is to be expected. However, the use of mental shortcuts, whereby complex problems are simplified to ease decision-

making, seems less plausible given the relatively large change in recommendation (e.g., a decrease in strongly agree to continue the programme of 17, 10 and 7 percentage points respectively (Fig 2). Another perspective might come from the theory of cognitive dissonance [56]: people confronted with knowledge that conflicts with their previously held opinions might dismiss the new information (or perceive the new information selectively) as a way of coping with unpleasant feelings arising from cognitive dissonance. Selective perception has previously been documented in relation to information about mammography screening [57]. In the presence of selective perception, we would expect that participants with high knowledge at recruitment (an indicator of awareness of harms) would be more inclined to change recommendation towards a more sceptical position compared to participants without knowledge of harms, which was the case in our study. Participants with high knowledge had almost five times the odds of changing towards a more sceptical position compared to participants with low knowledge (S2 Table). In the present study, we cannot determine if cognitive dissonance affected recommendations about mammography screening. However, striving towards consistency seemed to affect participants' opinions: the proportion of participants who made a recommendation in accordance with what they stated as an acceptable mortality reduction and an acceptable amount of overdiagnosis almost doubled (from 21% to 40%) (S3 Table).

Contrary to cognitive theories, extensive literature has dealt with opinions as something evolving through experiences, behaviour and social interactions. Looking at different subgroups of our participants in relation to support for the programme at recruitment (Table 2), people working within the healthcare sector, women, and people in the screening age groups (51–60 years and 61–70 years) were the subgroups with the highest proportion of participants in strong support of mammography screening; 85%, 79%, 83% and 87% respectively. People in these subgroups were most likely to have participated in or been in contact with one of the three cancer screening programmes in Denmark. The pattern might indicate that peoples' behaviour has a reciprocal effect on their immediate recommendation [58]. However, people with breast cancer in the family, women, and those belonging to the screening age group were not statistically significantly more or less inclined to change recommendation towards a more sceptical position (S2 Table), which could indicate that these parameters do not affect opinion change. The findings could also be due to low statistical power.

According to sociological research, participation in screening relates to ideas of self-responsibility and social obligation [59]. Opinions about mammography screening in society might well be motivated by these same ideas. However, the relative importance of the opinion that 'it is mandatory to participate in mammography screening' for predicting recommendation was small (between 1% and 6%) (S1 Fig).

Our data cannot explain why the change in knowledge and recommendation change are not correlated but our data dismiss an instrumental understanding of knowledge as the basis for reaching a recommendation about mammography screening in the population. However, knowledge still played a part in opinion formation: a high level of knowledge at recruitment was correlated to an increased probability of changing recommendation towards a more sceptical position OR 4.83 (1.40;16.65) (S2 Table).

A change in the overall recommendation and its lack of relationship with acquired knowledge suggests that despite the recommendations being based on a higher level of understanding, there are still many other factors at play. For example, the fear of the disease. This implies a clash between emotions and rationality.

The deliberative polling process affected participants in other ways than making them more sceptical towards mammography screening. In general, participants' recommendations became anchored in more opinion dimensions because of the process of information and deliberation. At recruitment ($T_1$) the explanatory efficacy of the two opinion items 'regret' and

'effect' was 50%, whereas more opinion items contributed to predicting the recommendation at later inquiry time points (S1 Fig). These results could suggest that participants became more nuanced as a result of the process. Participant 'worry' decreased in relative importance for predicting recommendation from recruitment throughout the process as did 'mandatory 1'–feeling it is mandatory to participate in mammography screening (for $T_2$ and $T_3$). These results might reflect the idea that participants shift towards basing their arguments on more general principles (as opposed to personal interest) because general arguments will be more convincing to others.

## Changes in preferences regarding mortality reduction and overdiagnosis

More participants accepted one avoided death per 1000 women over a 20-year screening period after information and deliberation compared with before; 42% at $T_3$ vs. 25% at $T_1$ (Fig 3). In addition, a shift towards acceptance of more overdiagnosis was seen from $T_1$ to $T_3$ (Fig 3). This was, however, not statistically significant. A key explanation could be that participants avoid dissonance (cognitive discomfort) by adjusting preferences for overdiagnosis and mortality reduction to fit their support for the programme. Another explanation could be that participants avoid social discomfort by adjusting preferences to fit their support to comply with the premise of the Deliberative Poll, in which benefits and harms are linked to recommendations by virtue of the information given in the video. Another important explanation could be that, at recruitment, participants did not know what overdiagnosis meant, which would place our study in line with results from an Australian study demonstrating approximate awareness of the concept of overdiagnosis in only 41% of the population [60].

## Previous research

Our results are consistent with results from a Citizens' Jury conducted in Spain where a majority (11 to 2) wanted mammography screening to continue for women aged 50–69 [35]. Our results also support findings from Australia [34] that considerations other than the balance between benefits and harms are at play when laypeople consider mammography screening. A change in opinion towards viewing the harms as carrying more weight than the benefits was correlated to a change in recommendation towards being more sceptical (correlation coefficient -0.38). However, this opinion change was of relatively little importance in relation to change in recommendation (RI 6%) (Table 3). In addition, 60% of participants made a recommendation that was not in accordance with their preferences regarding mortality reduction and overdiagnosis (S3 Table, after information and deliberation 40% made a recommendation that was in accordance with their preferences regarding mortality reduction and overdiagnosis).

A recent systematic review of cancer screening guidelines found that only 12% of the included guidelines commented on laypeople's preferences. In addition, none of the included guidelines defined a threshold for the key benefits that would be required to recommend screening, given the harms [61]. Our study shows that people are willing to state their preferences regarding mortality reduction and overdiagnosis and, not surprisingly, there was a large variation in their preferences (Fig 3).

## Strength and limitations

A strength of the Deliberative Poll design, compared to a Citizens' Jury, relates to the size and the generalisability of recommendations. Where the Citizens' Jury includes a limited number of participants (11–18 each), Deliberative Polls include more people (in the present study, 89 participants) and careful demographic selection can ensure that participants are representative

of the general population. Whereas the juries are well suited for qualitative analysis, exploring different underlying values and reasons behind a given recommendation, the quantitative design of the Deliberative Poll allows for statistical analysis of participants' (gain in) knowledge about the topic and how it develops with the opinions they expressed before, during, and after the Deliberative Poll. In addition, our design allowed for analysis of changes in the degree of support for the mammography screening programme which a voting procedure (dichotome recommendation -for or against) does not capture.

Our Deliberative Poll included both men and women in the discussions about mammography screening in contrast to most citizen jury research where men are excluded. Including a "mini-public" in the Deliberative Poll emphasises two democratic ideals: first that decisions should be based on informed discussions and second, that people affected by the decision should be included in the discussions. Screening programmes like mammography screening could be argued to impact all of society, not only women in the screening age group. In a publicly funded healthcare system, as in Denmark, decisions about whether to provide a healthcare service involve trade-offs against other services. In addition, screening involves ethical considerations e.g. can we as a society accept a programme that benefits some at the expense of others? Excluding men from our Deliberative Poll or only including women in the screening age group might have resulted in different findings. As previously described, we found that women, and especially women in the screening age groups, were the subgroups with the highest proportion of participants in strong support of mammography screening at recruitment. Breast cancer is rarely a direct threat to men's lives which could be argued to affect their views on the programme. However, arguing that people should only take part in debates about topics affecting their own lives directly seems limited.

In the Deliberative Poll on mammography screening differences in pre- and post-test of the participants are interpreted as caused by the deliberative polling process. In our opinion, the risk of effects from extraneous variables is minimal as mammography screening to our knowledge was not a topic for public discussion during the study period. However, to exclude a possible effect of extraneous variables pre- and post-test, a control group could have been added to the design. In addition, we cannot exclude a retest-bias discussed in our previous article [42]. Comparing (post-test) results between a control group tested several times and one only tested once could have been used to assess any potential retest effects [42]. Also, it would have been interesting to interview the same citizens before and after the video to see if they understood the questions in the same way.

The Deliberative Poll on mammography screening was originally planned as a face-to-face citizens' assembly lasting a weekend. Because of Covid-19 the assembly was conducted online and was shortened to prevent "online fatigue". Participants therefore did not have as long a time as planned to delve into the complex topic of screening. Also, the online modality meant that some people were excluded during recruitment because of their lack of technical equipment. As described in our previous article, online rehearsals were offered the week before the assembly in an attempt to minimise selection bias due to the online modality.

Our sample size calculation advised inclusion of at least 100 participants, but, again due to the Covid-19 pandemic, we were only able to include 89. We may not have had enough power, therefore, to detect statistically significant differences within our secondary analysis. The sample was statistically representative of the adult Danish population on sociodemographic parameters as well as on attitudes to and knowledge of mammography screening. However, for crossed cells the number of participants was too small to reach representativeness (e.g. the proportion of more highly educated women, aged 31–40 years, living in Region Zealand does not correspond to the proportion in society). A larger sample size could have ensured cross-cell representativeness and prevented possible biases from emerging if subgroups reacted

differently to the deliberative polling process. Just under 2000 people were contacted at the beginning of the recruitment process. Of those, 89 people were included in the study (Fig 1). A large proportion of people did not want to participate or did not respond to the emailed survey. A possible volunteer bias is therefore impossible to rule out.

## Conclusion

The deliberative polling process made our lay participants less supportive of the mammography screening programme: 72% were in strong support of continuation at recruitment, whereas 55% were in strong support after information, and 65% after deliberation. The decrease in support was not correlated to an increase in knowledge. While the balance between benefits and harms has been central in discussions about mammography screening within the medical professions, other issues such as viewing the programme as mandatory were better at predicting participants' support of the programme compared to how they viewed the balance between benefits and harms. However, the balance between benefits and harms increased in importance relative to other issues as a result of the deliberative polling process. Participants' recommendations about the continuation of mammography screening did not, to a large extent, reflect their preferences regarding mortality reduction and overdiagnosis in the programme. Nevertheless, the proportion of participants who made a recommendation following what they stated were acceptable rates of mortality reduction and with what they stated were acceptable levels of overdiagnosis rose after information and deliberation compared with at the time of recruitment. There was a tendency for people with a low level of knowledge about overdiagnosis to be more inclined to accept high levels of overdiagnosis (100,150, and 500 overdiagnosis per 1000 women invited to screening in 20 years). Accepting high levels of overdiagnosis increased the likelihood of being in strong support of the programme.

### Implications for policy

Policymakers must be more critical in their appraisal of the mammography screening programme listening to the considered lay recommendation compared with the immediate lay recommendation.

### Implications for research

One of many explanations as to why participants' recommendations do not follow their preferences is that mammography screening has already been implemented and is widely accepted in Danish society. It would be interesting to see if the same pattern was present in a Deliberative Poll about potential future screening programmes, e.g. abdominal aortic aneurism screening or low-dose CT screening for lung cancer. In addition, it would be interesting to explore in a qualitative study the reasons behind changes in participants' preferences: acceptance of less mortality reduction and more overdiagnosis after information and deliberation.

### Supporting information

**S1 Fig. Relative importance of opinion variables for predicting the recommendation to continue with mammography screening at four inquiry time points.** Note: Relative importance calculated in dominance analysis. Upper left: $T_1$ = inquiry time point 1 (recruitment) $R^2$ = 27.7, Upper right: $T_2$ = inquiry time point 2 (after video information) $R^2$ = 59.7, Lower left: $T_3$ = inquiry time point 3 (after deliberation) $R^2$ = 52.4, Lower right: $T_4$ = inquiry time point 4 (one month after the Citizens' Assembly) $R^2$ = 51.5.
(TIFF)

**S2 Fig. The relations between knowledge about overdiagnosis, acceptance of overdiagnosis and recommendation after information and deliberation ($T_3$).** Note: Odds Ratio = OR (95% CI). Low level of knowledge about overdiagnosis was defined as answering zero one or two out of three knowledge questions correctly as opposed to answering all three questions correctly (high level of knowledge). Acceptance of much overdiagnosis was defined as accepting 100, 150 or 500 overdiagnosis per 1000 women invited for screening in 20 years as opposed to accepting low level of overdiagnosis (zero, one or five overdiagnosis per 1000 women invited over 20 years).
(TIFF)

**S1 Table. Level of knowledge (% correct answers).** Note: The table shows the level of knowledge at the four poll inquiry time points expressed as % correct answers to the 13 knowledge items. $T_1$ = inquiry time point 1 (recruitment), $T_2$ = inquiry time point 2 (after video information), $T_3$ = inquiry time point 3 (after deliberation), $T_4$ = inquiry time point 4 (one month after the citizens' assembly).
(TIFF)

**S2 Table. Likelihood of change in recommendation towards a more sceptical position related to knowledge at recruitment and sociodemographic parameters.** Note: OR = Odds Ratio. Knowledge was divided according to the proportion of correct answers to 13 knowledge questions.
(TIFF)

**S3 Table. Recommendations in accordance with preferences regarding mortality reduction and overdiagnosis.** Note: The table combines the answers to three questions. 1. Participants recommendation (agree or disagree to continue mammography screening in Denmark). 2. What participants consider to be an acceptable rate of mortality reduction in such a programme. 3. What participants consider to be an acceptable level of overdiagnosis in the programme. A preference matching with a recommendation to continue was defined as acceptance of one, two or five avoided deaths per 1000 women invited for mammography screening for 20 years + acceptance of 20, 100, 150 or 500 overdiagnoses per 1000 women invited in 20 years. A preference matching with a recommendation not to continue was defined as acceptance of 20 or 50 avoided deaths per 1000 women invited for mammography screening for 20 years + acceptance of zero, one or five overdiagnoses per 1000 women invited over 20 years.
(TIFF)

## Author Contributions

**Conceptualization:** Manja D. Jensen, Kasper M. Hansen, Volkert Siersma, John Brodersen.

**Data curation:** Manja D. Jensen, Kasper M. Hansen, Volkert Siersma.

**Formal analysis:** Manja D. Jensen, Volkert Siersma.

**Funding acquisition:** Manja D. Jensen, John Brodersen.

**Investigation:** Manja D. Jensen, Kasper M. Hansen, John Brodersen.

**Methodology:** Kasper M. Hansen.

**Project administration:** Manja D. Jensen.

**Software:** Manja D. Jensen, Volkert Siersma.

**Supervision:** Kasper M. Hansen, Volkert Siersma, John Brodersen.

**Validation:** Volkert Siersma.

**Visualization:** Manja D. Jensen, Kasper M. Hansen, Volkert Siersma, John Brodersen.

**Writing – original draft:** Manja D. Jensen.

**Writing – review & editing:** Manja D. Jensen, Kasper M. Hansen, Volkert Siersma, John Brodersen.

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
