## [Decision Letter · Decision Letter 0]

16 Feb 2023

PONE-D-22-34061Mammography screening: eliciting the voices of informed citizensPLOS ONE

Dear Dr. Jensen,

Thank you for submitting your manuscript to PLOS ONE. After careful consideration, we feel that it has merit but does not fully meet PLOS ONE’s publication criteria as it currently stands. Therefore, we invite you to submit a revised version of the manuscript that addresses the points raised during the review process.

We look forward to receiving your revised manuscript.

Kind regards,

Felix G. Rebitschek

Academic Editor

PLOS ONE

Journal Requirements:

Additional Editor Comments:

Dear authors,

Thank you very much for your contribution! I ask you to base your efforts significantly on the comments by Reviewer 1 and look forward to your revision.

Sincerely,

Felix Rebitschek

Reviewers' comments:

Reviewer's Responses to Questions

**Comments to the Author**

1. Is the manuscript technically sound, and do the data support the conclusions?

Reviewer #1: Partly

Reviewer #2: Yes

Reviewer #3: Yes

2. Has the statistical analysis been performed appropriately and rigorously? 

Reviewer #1: Yes

Reviewer #2: Yes

Reviewer #3: I Don't Know

3. Have the authors made all data underlying the findings in their manuscript fully available?

Reviewer #1: Yes

Reviewer #2: Yes

Reviewer #3: Yes

4. Is the manuscript presented in an intelligible fashion and written in standard English?

Reviewer #1: Yes

Reviewer #2: Yes

Reviewer #3: Yes

5. Review Comments to the Author

Reviewer #1: The paper addresses an important topic, public engagement in decision-making about mammography screening. The intervention study is elaborate and yields relevant results about the preferences of informed laypeople regarding mortality reduction and overdiagnosis, the support for the continuation of mammography screening program in Denmark and the changes in knowledge after a deliberative polling intervention. However, in my view the paper needs some revisions regarding the presentation of the study rationale, method and some results:

1) One major point is that the intervention study has already been published in 2021 in previous paper in PLOS One (https://doi.org/10.1371/journal.pone.0258869 ). The results in the present paper were not previously published, and also all figures and tables are original, this is all fine. However, the descriptions of the rationale of the study and most importantly of the intervention are too short in the present paper. E.g., the contents of the video and its duration are not described, the procedure of the deliberative polling day and the dependent measures are only very briefly summarized. The fact that the participants also received a fact sheet in addition to the video is only mentioned in the 2001 paper, not in the present paper. The authors refer to the 2001 paper several times in the methods section (p. 5ff.), however in my view also the present paper needs more details on the intervention and the measures to be interpretable on its own. As the results are being interpreted as effects of the intervention, the intervention and dependent variables need to be explained more precisely in the present paper.

2) The introduction section is relatively short and has some gaps; in my view, the following aspects need to be added: a) a definition of deliberative polling (should be moved from p. 5 in the Method section to the Introduction), b) a paragraph on prior research about how to communicate evidence about mammography screening, c) a summary of prior research with regard to hypotheses 2 and 3. Also, the introduction does not refer to the study of the authors from 2021. Some relevant results of the study are currently presented in the methods section (p. 7) and should be moved to the introduction.

3) Results I: Effect sizes should be described for all statistical tests, which is currently not the case.

4) Results II: Table 2 was partly unclear to me and should be clarified. E.g., do the correlation coefficients for the opinion items reflect correlations with change in recommendation or with change in knowledge (see title of table 2)? Also, I believe that the analysis of relative importance needs its own table or at least additional information; the percentage of relative importance with regard to the determination coefficients cannot be fully interpreted without showing the determination coefficients (and more information about these multiple regressions should be made available in the supplemental materials).

5) Discussion I: Effect sizes should be taken into account when discussing the effects.

6) Discussion II: The discussion focuses on the unexpected result for H2. Some of the alternative explanations might be plausible for H2, but seem less plausible given that H1 was confirmed at the same time (e.g. if participants used gut-feelings instead of reflecting on knowledge (p. 12, line 23), why did still many of them change their recommendation?). Thus, I would recommend to reflect more on the combination of the results and also to elaborate a bit more on the results of the other hypotheses. The discussion of H2 could also be shortened a bit by moving some of the new results presented here to the results section.

7) Discussion III: The section about limitations should be extended by the following points: What are the implications of a) the lack of a control group and b) the decision to use a representative sample of men and women, but not a representative sample of women only? Most previous studies studied women’s perceptions of mammography screening.

8) Conclusion: The conclusion currently reflects only parts of the main findings of the study with regard to the three hypotheses. Lines 11/12 are not clear to me – why does the balance between benefits and harms seem less important to the recommendations of informed lay people? The results of the study pointed into another direction.

Minor points:

- Hypothesis 2 is sometimes about knowledge gain (p. 4, 12), sometimes about change in knowledge (p. 9, Table 2), which is not necessarily equivalent. It would be helpful for readers if one of the two expressions would be used consistently when discussing H2.

- Descriptive data about the changes in knowledge, as well as absolute levels of knowledge, is currently missing and could be added as a table in the supplementary materials. This could also be helpful for interpreting the results of H2.

- Conclusion: The framing of the result of H3 is interesting: “Participants’ recommendations about the continuation of mammography screening did not, to a large extent, reflect their preferences regarding mortality reduction and overdiagnosis in the programme.” (p. 16) vs. “In line with our third hypothesis, recommendations were more in line with preferences regarding mortality reduction and overdiagnosis after information and deliberation compared with before.” (p. 12). Of course, both statements are correct. But which is the main point that should be included in the abstract and in the conclusions? Personally, I would prefer that both aspects are presented equitably.

- P. 3, 22-24: Some more current studies should be added.

- References [23] and [34] are very general and should be replaced with more specific references that are a bit closer to the present context of medical decision-making.

- A reference is missing for the theory of cognitive dissonance (p. 13, line 3).

- S1_Fig.: I would recommend to use bar charts instead of pie charts.

- S2_Table: Absolute frequencies should be added (e.g., number of participants educated or working in the healthcare sector).

Reviewer #2: Very interesting findings and a complex intervention!

One question: On page 9 you write: "Learning (change in knowledge index) was not correlated to a change in recommendation, correlation 0.03 (95%CI -0.19-12 0.24) (Table 2). On the contrary, a change in opinion towards viewing the harms of screening carrying more weight than the benefits, was correlated to a change in recommendation towards being more skeptical learning was not correlated to a change in recommendation." In our experience (not research!) this is not an contradiction as information (e.g. statistical facts) has very different meanings for individuals. That is how decisions aids work (give information + als "what does this mean fo me?)

Maybe ist is possible to summarise in short (e.g. in the abstract) what made people change their recommendation (if it is not knowlege and if the opion items had relatively little importance).

Reviewer #3: Thank you for the opportunity to review the paper „Mammography screening: eliciting the voices of informed citizens“

Aim of the present study was to examine the public’s recommendations about the continuation of mammography screening and the preferences regarding acceptable rates of mortality reduction and overdiagnosis. For the public involvement, an online Deliberative Poll was held in Denmark in 2020 and outcoemes were assessed on four time points. Results on knowledge and opinion items have already been published in PLoS One in 2021 (https://doi.org/10.1371/journal.pone.0258869).

This publication is referenced in the present manuscript (reference 31) in order to refer to the methodological approach as well as previous results. However, these references do not make it clear that the present manuscript is a secondary analysis, or at least a further analysis of the data already published. If it seems appropriate to submit more than one publication for a study, the authors should describe and justify this procedure transparently.

Apart from that, the subject is important and the method for public involvement is innovative. The results are relevant and could form the basis for further research into the relationship between knowledge, attitudes and recommendations for screening. Knowledge on these associations (or the lack of them) could also be important for interventions to enhance individual informed decisions.

In the following, find some smaller comments on the manuscript and supplementing materials:

- Page 15, lines 12 ff: The attribution of the descriptions in the text to the items and results in Table 2 is not entirely clear. Possibly the effects could be represented here, as in line 11.

- Table S2 is introduced only in the discussion. This is rather unusual, and it would be better to take up only results that have already been presented in advance in the discussion.

- The arrows and numbers in Fig. 1 should be explained in the legend.

- Fig. 2: To avoid framing of data, the axes should always go up to 100 in the representation of percent.

- Fig. S1: In the current display, the assignment of the graphics to the times is missing.

6. PLOS authors have the option to publish the peer review history of their article (what does this mean?). If published, this will include your full peer review and any attached files.

Reviewer #1: No

Reviewer #2: **Yes: **Ulla Sladek

Reviewer #3: No

---

## [Author Response · Author response to Decision Letter 0]

21 Jul 2023

Response to Reviewers

Journal Requirements:

In the revised manuscript we have moved the quote by Joubert to page two to make it clear that the quote is not part of the title. To the best of our knowledge the manuscript now meets PLOS ONE's style requirements.

In the revised manuscript we have deleted the text “insert Fig.1.” and “insert Fig.2.”. In order to help with the layout we have uploaded a separate file called “Layout Fig1.and Fig2.” that shows how Fig1. and Fig2 can be formatted in relation to the text in the figure (heading and note).

Information is now added about informed consent in the ethics statement in the methods section and online. Our study did not include minors.

Review Comments to the Author

Reviewer #1: 

The paper addresses an important topic, public engagement in decision-making about mammography screening. The intervention study is elaborate and yields relevant results about the preferences of informed laypeople regarding mortality reduction and overdiagnosis, the support for the continuation of mammography screening program in Denmark and the changes in knowledge after a deliberative polling intervention. However, in my view the paper needs some revisions regarding the presentation of the study rationale, method and some results:

1) One major point is that the intervention study has already been published in 2021 in previous paper in PLOS One (https://doi.org/10.1371/journal.pone.0258869 ). The results in the present paper were not previously published, and also all figures and tables are original, this is all fine. However, the descriptions of the rationale of the study and most importantly of the intervention are too short in the present paper. E.g., the contents of the video and its duration are not described, the procedure of the deliberative polling day and the dependent measures are only very briefly summarized. The fact that the participants also received a fact sheet in addition to the video is only mentioned in the 2001 paper, not in the present paper. The authors refer to the 2001 paper several times in the methods section (p. 5ff.), however in my view also the present paper needs more details on the intervention and the measures to be interpretable on its own. As the results are being interpreted as effects of the intervention, the intervention and dependent variables need to be explained more precisely in the present paper.

In the revised manuscript more information about the content of the video is added in the section ”The Deliberative Polling process”. In the same section we have also added that participants received a fact sheet as well as more information about the deliberative procedure and the dependent measures.

2) The introduction section is relatively short and has some gaps; in my view, the following aspects need to be added: 

a) a definition of deliberative polling (should be moved from p. 5 in the Method section to the Introduction). 

b) a paragraph on prior research about how to communicate evidence about mammography screening, 

c) a summary of prior research with regard to hypotheses 2 and 3. Also, the introduction does not refer to the study of the authors from 2021. Some relevant results of the study are currently presented in the methods section (p. 7) and should be moved to the introduction.

 In the revised manuscript we have moved the description of a Deliberative Poll from the methods section to the introduction. In addition, we elaborate more on the Deliberative Poll and the rationale of this method in the methods section. 

In the revised manuscript we have included a paragraph on prior research about how to communicate evidence about mammography screening (Introduction section). 

In the revised manuscript we describe and refer to our study from 2021 in the introduction section. In addition, the results described in the methods section is now moved to the introduction.

3) Results I: Effect sizes should be described for all statistical tests, which is currently not the case.

For analysis besides the chi-squared tests we present correlation coefficients and OR’s. In the revised manuscript we have added percentage point difference to the chi-squared/t-tests.

4) Results II: Table 2 was partly unclear to me and should be clarified. E.g., do the correlation coefficients for the opinion items reflect correlations with change in recommendation or with change in knowledge (see title of table 2)? Also, I believe that the analysis of relative importance needs its own table or at least additional information; the percentage of relative importance with regard to the determination coefficients cannot be fully interpreted without showing the determination coefficients (and more information about these multiple regressions should be made available in the supplemental materials).

In the revised manuscript we have specified in the header for the correlation coefficient (Table 2) that this is the coefficient for change in recommendation and change in knowledge score/item, and change in recommendation and change in opinion item, respectively. Similarly, for the headers for relative importance we have specified that these are the relative importance of the various changes in knowledge items for the change in recommendation, and the relative importance of the various changes in opinion items for the change in recommendation respectively.

It is not possible to give the RI analysis their own table as the calculation of RI entail many regression analysis (2 to the power #factorrs =8192 for the 13 knowledge items.) Instead we have added the R-squared of the full model in Table 2 (see table note) as well as S1_Fig. (see table note).

5) Discussion I: Effect sizes should be taken into account when discussing the effects.

In the revised manuscript we have included effect sizes in the discussion section.

6) Discussion II: The discussion focuses on the unexpected result for H2. Some of the alternative explanations might be plausible for H2, but seem less plausible given that H1 was confirmed at the same time (e.g. if participants used gut-feelings instead of reflecting on knowledge (p. 12, line 23), why did still many of them change their recommendation?). Thus, I would recommend to reflect more on the combination of the results and also to elaborate a bit more on the results of the other hypotheses. The discussion of H2 could also be shortened a bit by moving some of the new results presented here to the results section.

 In the revised manuscript we have shortened the discussion of H2 a bit. Reflecting on H1 and H2 in combination, we agree with the reviewer and in the revised manuscript we have included a comment in relation to the reviewers point; finding it less likely that participants used gut feelings when many changed their recommendation. 

7) Discussion III: The section about limitations should be extended by the following points: What are the implications of a) the lack of a control group and b) the decision to use a representative sample of men and women, but not a representative sample of women only? Most previous studies studied women’s perceptions of mammography screening.

In the revised manuscript (Strength and limitations section) we have included a discussion about the lack of a control group and our decision to use a representative sample (men and women) and not only woman.

8) Conclusion: The conclusion currently reflects only parts of the main findings of the study with regard to the three hypotheses. Lines 11/12 are not clear to me – why does the balance between benefits and harms seem less important to the recommendations of informed lay people? The results of the study pointed into another direction.

In the revised manuscript our conclusions regarding all three hypotheses are described. In addition, we have reframed the bit about the balance between benefits and harms.

Minor points:

9) Hypothesis 2 is sometimes about knowledge gain (p. 4, 12), sometimes about change in knowledge (p. 9, Table 2), which is not necessarily equivalent. It would be helpful for readers if one of the two expressions would be used consistently when discussing H2.

In the revised manuscript we use only “change in knowledge”.

10) Descriptive data about the changes in knowledge, as well as absolute levels of knowledge, is currently missing and could be added as a table in the supplementary materials. This could also be helpful for interpreting the results of H2.

In the revised manuscript we describe in the introduction section that participants’ knowledge about mammography screening increased markedly as a result of the Deliberative Polling process “at recruitment only 1% of participants were able to answer two-thirds of the knowledge questions correctly. After video information the proportion was 56% (With reference to our previous paper).

In the revised manuscript we have also included the supplementary table “Level of knowledge”. NB: this is a reprint of the table we published in our previous article Jensen MD, Hansen KM, Siersma V, J B. Using a Deliberative Poll on breast cancer screening to assess and improve the decision quality of laypeople. PloS one. 2021;16(10). We write in the table note that this is a reprint. Is it okay to publish a reprint like this?

11) Conclusion: The framing of the result of H3 is interesting: “Participants’ recommendations about the continuation of mammography screening did not, to a large extent, reflect their preferences regarding mortality reduction and overdiagnosis in the programme.” (p. 16) vs. “In line with our third hypothesis, recommendations were more in line with preferences regarding mortality reduction and overdiagnosis after information and deliberation compared with before.” (p. 12). Of course, both statements are correct. But which is the main point that should be included in the abstract and in the conclusions? Personally, I would prefer that both aspects are presented equitably.

In the revised manuscript both aspects are presented in the abstract as well as the conclusion.

12) P. 3, 22-24: Some more current studies should be added.

In the revised manuscript we have added the following studies: 

Yu J, Nagler RH, Fowler EF, Kerlikowske K, Gollust SE. Women’s Awareness and Perceived Importance of the Harms and Benefits of Mammography Screening: Results From a 2016 National Survey. JAMA internal medicine. 2017;177(9):1381-2.

Abelson J, Tripp L, Brouwers MC, Pond G, Sussman J. Uncertain times: A survey of Canadian women's perspectives toward mammography screening. Preventive Medicine. 2018;112:209-15.

13) References [23] and [34] are very general and should be replaced with more specific references that are a bit closer to the present context of medical decision-making.

In the revised manuscript we have deleted the following “general” bit and the referance: “This also applies to the public’s decision-making in a more general sense (Ref. 23: Tversky A, Kahneman D. The framing of decisions and the psychology of choice. Science. 1981;211(4481):453-8.)

14) A reference is missing for the theory of cognitive dissonance (p. 13, line 3)

In the revised manuscript the reference is added.

15) S1_Fig.: I would recommend to use bar charts instead of pie charts.

Because the relative importance adds up to 100% we find that pie charts to be natural.

16) S2_Table: Absolute frequencies should be added (e.g., number of participants educated or working in the healthcare sector).

In the revised manuscript absolute frequencies are added to the table.

Reviewer #2: 

17) Very interesting findings and a complex intervention!

One question: On page 9 you write: "Learning (change in knowledge index) was not correlated to a change in recommendation, correlation 0.03 (95%CI -0.19-12 0.24) (Table 2). On the contrary, a change in opinion towards viewing the harms of screening carrying more weight than the benefits, was correlated to a change in recommendation towards being more skeptical learning was not correlated to a change in recommendation." In our experience (not research!) this is not an contradiction as information (e.g. statistical facts) has very different meanings for individuals. That is how decisions aids work (give information + als "what does this mean fo me?)

Maybe ist is possible to summarise in short (e.g. in the abstract) what made people change their recommendation (if it is not knowlege and if the opinion items had relatively little importance).

Thank you for the comment. Unfortunately, we can only hypothesize on this based on our results and study design and therefore we only discuss this in our discussion section. We agree that it could be interesting to do qualitative work in relation to a Deliberative Poll which could probably shed more light on what made people change their recommendation.

Reviewer #3:

Thank you for the opportunity to review the paper „Mammography screening: eliciting the voices of informed citizens“

Aim of the present study was to examine the public’s recommendations about the continuation of mammography screening and the preferences regarding acceptable rates of mortality reduction and overdiagnosis. 

18) For the public involvement, an online Deliberative Poll was held in Denmark in 2020 and outcoemes were assessed on four time points. Results on knowledge and opinion items have already been published in PLoS One in 2021. This publication is referenced in the present manuscript (reference 31) in order to refer to the methodological approach as well as previous results. However, these references do not make it clear that the present manuscript is a secondary analysis, or at least a further analysis of the data already published. If it seems appropriate to submit more than one publication for a study, the authors should describe and justify this procedure transparently.

Thank you for the good comment. In the revised version of the manuscript we have included information about our previous study in the introduction section (the aim and main results) just before the aim of the present study. We hope this contribute to transparency. The aim in the present paper is very different from the aim in the first paper and the results in the present paper is not previously published.

Apart from that, the subject is important and the method for public involvement is innovative. The results are relevant and could form the basis for further research into the relationship between knowledge, attitudes and recommendations for screening. Knowledge on these associations (or the lack of them) could also be important for interventions to enhance individual informed decisions.

In the following, find some smaller comments on the manuscript and supplementing materials:

19) Page 15, lines 12 ff: The attribution of the descriptions in the text to the items and results in Table 2 is not entirely clear. Possibly the effects could be represented here, as in line 11.

In the revised manuscript we have added the following text “(S1 Table, after information and deliberation 40% made a recommendation that was in accordance with their preferences regarding mortality reduction and overdiagnosis”) in order to link the text about 60% (100-40%) to the S1 Table.

20) Table S2 is introduced only in the discussion. This is rather unusual, and it would be better to take up only results that have already been presented in advance in the discussion.

In the revised manuscript res

---

## [Decision Letter · Decision Letter 1]

17 Oct 2023

PONE-D-22-34061R1Mammography screening: eliciting the voices of informed citizensPLOS ONE

Dear Dr. Jensen,

Thank you for submitting your manuscript to PLOS ONE. After careful consideration, we feel that it has merit but does not fully meet PLOS ONE’s publication criteria as it currently stands. Therefore, we invite you to submit a revised version of the manuscript that addresses the points raised during the review process.

In the light of earlier revising and addressing the points of the reviewers, it may be another effort to process the comments below. But major concerns need to be addressed for continuing with the manuscript. Some of them may be solved by just better explaining the details or the thoughts behind.==============================

We look forward to receiving your revised manuscript.

Kind regards,

Felix G. Rebitschek

Academic Editor

PLOS ONE

Additional Editor Comments:

Majors

• What is the research question?

o Do the authors want to inform about potential advantages and effects by the specific method Deliberative Poll? Is it a paper about a method? Then, more justification is needed about why selecting the mammography case?

o Do the authors want to inform an issue in breast cancer guideline development, because the study reveals neglected attitudes of informed citizens that from now on can be used? Is it a paper about qualitative insights of mammography beliefs and attitudes in a Western population?

• What is a public’s recommendation? What is the role of the citizens exactly? Why they give recommendations to whom – is there any real-world equivalent?

o What is the exact subject of the study – an age-stratified population-based mammography screening without risk assessment? Is it a an offer, obligatory, biannualy, yearly, age-range? This clearly affects the interpretation of the study – I could read it off from figure/table captions, but this is too central.

• What is the argument for applying the method? This needs much more argumentation

o “[other methods’] limited representativeness of the wider population” – with 100 people the authors do not reach population representativeness, so what is meant by that (please define in which way the author’s sample is representative)? The generalisation of a convenience sample is negligible if it is about attitudes. Is the point about a stratified sampling to assure that certain vulnerable subgroups are included? How do the authors show the “representativeness” advantage?

o “range of arguments” is also unclear. There is some plausibility that the authors reveal more aspects when they do qualitative research with 100 instead of 10 participants, but why not 20 or 30? What makes this method reveal sufficiently broader arguments and do the authors have evidence for that (references)? Particularly in the light of so much interesting qualitative research with 10-15 participants… Later I found out that you do not provide qualitative insights, so we are talking about a cohort survey study.

• The third hypothesis requires more background, because the authors want to compare two flying bullets: a changing recommendation-related judgment (an expression of preference) and an evaluation of a required minimum benefit and maximum harm. The authors here touch the fields of preference elicitation and shared decision-making and should provide more conceptual ground on the key-strength of the described study: linking option knowledge with underlying preferences and final intentions/recommendations. What was done in the supplementary figure on overdiagnoses should have be done on the full knowledge, the most aggregated preference assessment that the authors have, with recommendation.

• Method section:

o With regard to your representativeness argument, information on recruitment needs to be extended here: sampling, frame, inclusion/exclusion criteria, participation rates, risk of a topical bias – how were candidates approached? Avoid the bold statements (p.7,l12-19; not only because of the age range 18-70 and the online study requirements, but also because in crossed cells the impressive alignment with the general Danish population (on the main factors) does not hold; even for crossed cells of age x education x gender 100 people are not sufficient)

o please restructure into distinct paragraphs for study material and procedure

o why the separate analyses for the knowledge on overdiagnosis and mortality reduction, but not on the other eleven items?

o Provide more information about the used measures here in a separate measures paragraph (for instance the preference assessments, such as acceptable overdiagnosis)

o Please provide the full items including false options for knowledge (beyond the supplementary reprint)

 What is the reference for the ground truth and for the range that you scored correct in knowledge, numerical?

 What is the reliability of this assessment?

• Results

o Proportions of recommendation statements should be accompanied by information about uncertainty around those estimates

o Fig. 2:

 I do not understand the mortality-reduction item and I doubt that laypeople could so. Is this a problem of an English translation?

 Have you binned the responses or was it multiple choice?

 The formatting needs to be checked against PLOSONE standards

 Please insert Confidence intervals or any other indication of uncertainty around the survey estimates.

o Table 1.

 Is it [%] in each column?

 A dominance analysis with such a small sample over so many items requires information about what is the uncertainty around the estimate of relative importance for each item

• Discussion:

o limitations by your method needs to more elaboration, starting with the generalisability of your findings to which population

o what is role of modality for deliberate polls – offline, hybrid, online?

o please elaborate on the methodological point of equal considerations of input by women among your participants (who are or will be offered mammography screening) and other participants (e.g., men) in determining a public’s consideration – within participatory research literature distinguishing those who are directly affected by a subject and those who are not

• Please rewrite the abstract which cannot be understood by a reader without the authors‘ background (what is a recommendation by laypeople for whom, type of mammography screening, what is the method, representativeness in which way, result focus on the method’s contribution)

• Language editing, please let a native lecturer go over the manuscript which contains many types of errors (e.g., p.3, l.14 incomprehensible; which evaluation?; p.4,l.22) and unusual expressions

Minors

• Please provide the correct permanent video link

• Why is public engagement in decision-making a way to address different preferences of experts?

• What are “community values”?

• What is opinion stability and opinion consistency – difference? References? Do we talk about attitudes?

• Please check for consistent capitalisation (different methods)?

• How could the people from the survey assemble physically – or was it fully online?

• What is “Citizen’s Juries”?

• How did you select “neutral” moderators?

• Please move derived hypothesis (p.9.l17-20) up to the introduction

• What does mean: “an informed recommendation is RELATED to preferences” ? informed decisions, for instance, are defined by an alignment of the benefit-harm evaluation of options, option knowledge and the chosen option (Theresa Marteau)

• Direct references for “Mammography screening is controversial”

• Check for an update of Mammography screening, e.g. Germany IQWIG recommended extension to new age groups

• Ref. for benefit overestimation worth to consider: Gigerenzer, G., Mata, J., & Frank, R. (2009). Public knowledge of benefits of breast and prostate cancer screening in Europe. Journal of the National Cancer Institute, 101(17), 1216-1220.

• Please provide a reference for P.6.,l. 20-22 – consider moving the sentence to the introduction of deliberate poll

• P.10, l.1-3. The sentence is unclear to me

• Is correlation 0.03 => r = .03 (please check results descriptions against APA/PLOSONE)

• Could you please restructure the supplementary table about recommendation-continuing: here I read 3 agree at t3 with “continue” although they fall under “preferences matching with a recommendation not to continue”…? unclear

Reviewers' comments:

Reviewer's Responses to Questions

**Comments to the Author**

1. If the authors have adequately addressed your comments raised in a previous round of review and you feel that this manuscript is now acceptable for publication, you may indicate that here to bypass the “Comments to the Author” section, enter your conflict of interest statement in the “Confidential to Editor” section, and submit your "Accept" recommendation.

Reviewer #1: All comments have been addressed

2. Is the manuscript technically sound, and do the data support the conclusions?

Reviewer #1: (No Response)

3. Has the statistical analysis been performed appropriately and rigorously? 

Reviewer #1: (No Response)

4. Have the authors made all data underlying the findings in their manuscript fully available?

Reviewer #1: (No Response)

5. Is the manuscript presented in an intelligible fashion and written in standard English?

Reviewer #1: (No Response)

6. Review Comments to the Author

Reviewer #1: (No Response)

7. PLOS authors have the option to publish the peer review history of their article (what does this mean?). If published, this will include your full peer review and any attached files.

Reviewer #1: No

---

## [Author Response · Author response to Decision Letter 1]

29 Mar 2024

PONE-D-22-34061R1

Mammography screening: eliciting the voices of informed citizens

PLOS ONE

Dear Dr. Jensen,

Thank you for submitting your manuscript to PLOS ONE. After careful consideration, we feel that it has merit but does not fully meet PLOS ONE’s publication criteria as it currently stands. Therefore, we invite you to submit a revised version of the manuscript that addresses the points raised during the review process.

In the light of earlier revising and addressing the points of the reviewers, it may be another effort to process the comments below. But major concerns need to be addressed for continuing with the manuscript. Some of them may be solved by just better explaining the details or the thoughts behind.

We look forward to receiving your revised manuscript.

Kind regards,

Felix G. Rebitschek

Academic Editor

PLOS ONE

Additional Editor Comments:

Majors

• What is the research question?

o Do the authors want to inform about potential advantages and effects by the specific method Deliberative Poll? Is it a paper about a method? Then, more justification is needed about why selecting the mammography case?

o Do the authors want to inform an issue in breast cancer guideline development, because the study reveals neglected attitudes of informed citizens that from now on can be used? Is it a paper about qualitative insights of mammography beliefs and attitudes in a Western population? 

Thank you for the comment. The focus of our first paper from the project (“Using a Deliberative Poll on breast cancer screening to assess and improve the decision quality of laypeople”, published 2021 in PLOS ONE) was whether or not our experimental setup – the Deliberative Poll - approached the purpose or potential of this type of citizens’ involvement in decision making regarding continuing the Danish mammography screening program. Ie.a stronger focus on the method of Deliberative Polling on this complex issue.

The present study has a more substantial focus. I.e. how information and deliberation affected citizens’ recommendations about continuing the Danish mammography screening program. The primary aim was to determine an informed public’s recommendations about the continuation of the program in Denmark as well as to determine whether an informed recommendation was related to preferences regarding mortality reduction and overdiagnosis. The secondary aim was to examine how the process of information and deliberation affected opinions related to mammography screening. This is now highlight and explained in the introduction to the paper.

• What is a public’s recommendation? What is the role of the citizens exactly? Why they give recommendations to whom – is there any real-world equivalent?

Thank you for the comment. Why, when and how citizens should be involved in policy matters related to screening can be debated. Studies like ours cannot prove or disprove political theories about citizens’ involvement - whether or not we should involve citizens in decision-making about mammography screening. The focus of this paper is how information and deliberation affected citizens’ recommendations about continuing the program. I.e. descriptively. We hope this is clearer in the revised manuscript. 

According to Degeling, Carter and Rychetnik in their paper “Which public and why deliberate? – A scoping review of public deliberation in public health and health policy research” the public is often constructed in one of three ways when they are engaged in healthcare matters; either as advocates, consumers or as citizens (the “pure” public). In the revised manuscript we reference this paper and give some examples of public involvement in Danish healthcare matters.

o What is the exact subject of the study – an age-stratified population-based mammography screening without risk assessment? Is it an offer, obligatory, biannualy, yearly, age-range? This clearly affects the interpretation of the study – I could read it off from figure/table captions, but this is too central. 

Thank you for the comment. In the revised manuscript we describe the Danish programme (the subject of the study) in the beginning of the introduction. 

• What is the argument for applying the method? This needs much more argumentation

o “[other methods’] limited representativeness of the wider population” – with 100 people the authors do not reach population representativeness, so what is meant by that (please define in which way the author’s sample is representative)? The generalisation of a convenience sample is negligible if it is about attitudes. Is the point about a stratified sampling to assure that certain vulnerable subgroups are included? How do the authors show the “representativeness” advantage?

The sample is never representative or not. It is always a matter of degree, but in our case, it is quite good. Both in terms on socio demographics (e.g., gender, age, education and marital status) and history of breast cancer our sample is not statically different from the population (see table 1). The sampling is drawn from a random selected panel (e.g., is not self-selected). Participants were incentivized to participate, and much effort was put into the recruitment. The recruitment and the participants’ characteristics is described in detailed in our method-paper published in PloS ONE in 2021 https://www.ncbi.nlm.nih.gov/pmc/articles/PMC8530304/pdf/pone.0258869.pdf. We now include some information on this and provide a direct reference to this paper. 

The “range of arguments” is also unclear. There is some plausibility that the authors reveal more aspects when they do qualitative research with 100 instead of 10 participants, but why not 20 or 30? What makes this method reveal sufficiently broader arguments and do the authors have evidence for that (references)? Particularly in the light of so much interesting qualitative research with 10-15 participants… Later I found out that you do not provide qualitative insights, so we are talking about a cohort survey study. 

We agree that there is much interesting qualitative research with 10-15 participants including the qualitative studies based on Citizens’ Juries about screening referenced in our manuscript. We did a survey study and in the revised manuscript the abstract now mentions “survey data” to highlight this in the beginning. Our aim of 100 participants was based on various power tests in term if the group of participants would be a large enough group to find relevant statistical differences.

• The third hypothesis requires more background, because the authors want to compare two flying bullets: a changing recommendation-related judgment (an expression of preference) and an evaluation of a required minimum benefit and maximum harm. The authors here touch the fields of preference elicitation and shared decision-making and should provide more conceptual ground on the key-strength of the described study: linking option knowledge with underlying preferences and final intentions/recommendations. What was done in the supplementary figure on overdiagnoses should have be done on the full knowledge, the most aggregated preference assessment that the authors have, with recommendation.

Thank you for the comment. In the revised manuscript (introduction section) we elaborate on how group deliberation is believed to affect inconsistent opinions which function as background for our third hypothesis. In the revised manuscript we refer to this information in the description of our third hypothesis

• Method section:

o with regard to your representativeness argument, information on recruitment needs to be extended here: sampling, frame, inclusion/exclusion criteria, participation rates, risk of a topical bias – how were candidates approached? Avoid the bold statements (p.7,l12-19; not only because of the age range 18-70 and the online study requirements, but also because in crossed cells the impressive alignment with the general Danish population (on the main factors) does not hold; even for crossed cells of age x education x gender 100 people are not sufficient).

Thank you for the comment. In the revised manuscript the recruitment process is described in more detail and a flow diagram for the recruitment is now added (Fig.1). The flow diagram includes the number of participants approached as well as excluded. There is also a reference to the method-paper mention above.

o please restructure into distinct paragraphs for study material and procedure

Thank you for the comment. In the revised manuscript we have restructured the methods section.

o why the separate analyses for the knowledge on overdiagnosis and mortality reduction, but not on the other eleven items? 

Thank you for the comment. We made a separate analysis concerning what we refer to as participants “preferences” regarding overdiagnosis and mortality reduction (Fig.4) and a separate analysis regarding the relations between knowledge about overdiagnosis, acceptance of overdiagnosis and recommendation after information and deliberation (T3) (S2.Fig). Mortality reduction and overdiagnosis are often highlighted as the most important benefit and harm of the program and we therefore found it interesting to look closer at these two aspects.

o Provide more information about the used measures here in a separate measures paragraph (for instance the preference assessments, such as acceptable overdiagnosis. 

Many of the survey questions were developed specifically for this project. We follow general survey methodology and try to minimize the cognitive demand on participants by applying classic Likert scales when every possible. Other items were developed and tested in previous Deliberative Polls (mostly the question on evaluation the deliberative and the briefing materials). The questionnaires were tested in a focus group before the distribution, and we also reviewed input from the Survey company who collected the data (Gallup). 

o Please provide the full items including false options for knowledge (beyond the supplementary reprint)

 What is the reference for the ground truth and for the range that you scored correct in knowledge, numerical?

 What is the reliability of this assessment?

Thank you for the comment. The full knowledge items are now presented in Fig.2. 

The information presented in the video and fact sheet functioned as “ground truth”. In the method section “Video information” we have references to the Cochrane report, The Danish Quality Database of Mammography Screening (Annual Report 2019), Statistics Denmark and an economic evaluation of the Danish programme. They were used as sources for mortality reduction, overdiagnosis, etc. presented in the video. In the revised manuscript we have given the information about the video in its section and the section “Data collection and trial outcome” we have added that the video information and fact sheet functioned as “ground truth” concerning knowledge.

We do not have a reference for the range we accepted. We choose to accept a range of numbers as correct because we acknowledge the uncertainty surrounding the numbers. See note in Table 1. for the range accepted as correct. We mention the uncertainty at the beginning of the video to be transparent about this. 

We could have made a test-retest (for example with 2-4 weeks span) in order to test the reliability. However, we know from psychometric research that items that are unambiguous, easy to understand and have high relevance have higher reliability compared to items with the opposite characteristics. In our study we aimed for high content validity and thereby indirect we tried to increase reliability.

• Results

o Proportions of recommendation statements should be accompanied by information about uncertainty around those estimates.

Our Fig.3 displays participants’ responses to the question “To what extent do you agree that we should continue mammography screening in Denmark” Responses (strongly agree, agree etc.) are shown in %. In the revised manuscript we have included 95% CI.

o Fig. 2:

 I do not understand the mortality-reduction item and I doubt that laypeople could so. Is this a problem of an English translation? 

Thank you for the question. We did three think-aloud tests (in Danish) to get an understanding of how lay individuals who had not viewed the mammography screening video understood our questions and we modified the wording accordingly. This information is added to the revised manuscript. 

We have adjusted the translation slightly, so it is truer to the actual question asked. Please also remember that the participants answer the question after a longer discussion on the issue.

 Have you binned the responses or was it multiple choice? 

Thank you for the comment. It was a multiple choice. In the revised manuscript this information is added to the figure text.

 The formatting needs to be checked against PLOSONE standards

Thank you for the comment. The manuscript no longer contains multiple columns concerning figure text.

 Please insert Confidence intervals or any other indication of uncertainty around the survey estimates.

The bar chart shows the percentage of participants accepting different thresholds for mortality reduction and different levels of overdiagnosis. In the revised manuscript we have included 95% CI.

o Table 1.

 Is it [%] in each column? 

Thank you for the comment. Yes. In the revised manuscript, we have added “%” to each column (beside % in the header).

 A dominance analysis with such a small sample over so many items requires information about what is the uncertainty around the estimate of relative importance for each item

While naïve uncertainty information, e.g. 95% confidence intervals on the estimates of relative importance, may be obtained by bootstrap, this will require quite some time investment, primarily to execute these calculations. Furthermore, there will be considerable trouble representing such uncertainty information in the figures in an understandable way, and trouble on how to interpret this information relative to pointing out the most important factors. We view inclusion of uncertainty estimates in the dominance analysis an (time-consuming) exercise that will cause much confusion and not lead to different conclusions.

• Discussion:

o limit

---

## [Editor Report · Decision Letter 2]

19 Apr 2024

PONE-D-22-34061R2Mammography screening: eliciting the voices of informed citizensPLOS ONE

Dear Dr. Jensen,

Thank you for submitting your manuscript to PLOS ONE. After careful consideration, we feel that it has merit but does not fully meet PLOS ONE’s publication criteria as it currently stands. Therefore, we invite you to submit a revised version of the manuscript that addresses the points raised during the review process. Please submit your revised manuscript by Jun 03 2024 11:59PM. If you will need more time than this to complete your revisions, please reply to this message or contact the journal office at plosone@plos.org. Please include the following items when submitting your revised manuscript:A rebuttal letter that responds to each point raised by the academic editor and reviewer(s). You should upload this letter as a separate file labeled 'Response to Reviewers'.A marked-up copy of your manuscript that highlights changes made to the original version. You should upload this as a separate file labeled 'Revised Manuscript with Track Changes'.An unmarked version of your revised paper without tracked changes. You should upload this as a separate file labeled 'Manuscript'.If applicable, we recommend that you deposit your laboratory protocols in protocols.io to enhance the reproducibility of your results. Protocols.io assigns your protocol its own identifier (DOI) so that it can be cited independently in the future. For instructions see: https://journals.plos.org/plosone/s/submission-guidelines#loc-laboratory-protocols. Additionally, PLOS ONE offers an option for publishing peer-reviewed Lab Protocol articles, which describe protocols hosted on protocols.io. Read more information on sharing protocols at https://plos.org/protocols?utm_medium=editorial-email&utm_source=authorletters&utm_campaign=protocols.

We look forward to receiving your revised manuscript.

Kind regards,

Felix G. Rebitschek

Academic Editor

PLOS ONE

Journal Requirements:

Additional Editor Comments:

Dear Authors,

Thank you very much for the extensive revision!

Only a few points remain:

- In Methods, under questionnaire, please include the item formulation used to assess the participants' view on what they considered to be an acceptable mortality reduction from mammography screening.

- Table 1 should follow the PLOSONE guidelines and the font size be equal to the manuscript text. Some response options reported in Table 1 are put in one line, others in separate lines.

- On P. 10 (without mark-ups), from lines 6-7, there is a break.

- Regarding my concern about the robustness of the dominance analyses, and in the light of the significance of the correlation coefficents for change in knowledge index and change in recommendation, it might be more comprehensible for the reader to not communicate importance for items with non-significant correlations

---

## [Author Response · Author response to Decision Letter 2]

27 Jun 2024

Response to reviewers

Journal Requirements:

Thank you for the comment. I have been through the reference list. WHO have moved their position paper to a new webpage which is now added to the reference list.

Additional Editor Comments:

Dear Authors,

Thank you very much for the extensive revision!

Only a few points remain:

- In Methods, under questionnaire, please include the item formulation used to assess the participants' view on what they considered to be an acceptable mortality reduction from mammography screening.

Thank you for the comment. In the revised manuscript we now mention the item regarding mortality reduction and the item regarding overdiagnosis in the methods section under questionnaire.

- Table 1 should follow the PLOSONE guidelines and the font size be equal to the manuscript text. Some response options reported in Table 1 are put in one line, others in separate lines.

Thank you for the comment. In the revised manuscript Table 1 have the same font size and each response option have a separate line.

- On P. 10 (without mark-ups), from lines 6-7, there is a break.

Thank you for the comment. There is no longer a break.

- Regarding my concern about the robustness of the dominance analyses, and in the light of the significance of the correlation coefficents for change in knowledge index and change in recommendation, it might be more comprehensible for the reader to not communicate importance for items with non-significant correlations

Thank you for the comment. Statistical significance is not the same as importance. Notably in small data sets, such as the one in the present paper, relatively large effects that may be considered important given the scenario at hand may find themselves not significant. Also, absence of evidence is not evidence of absence; but the reviewer/editor knows this. The findings in Table 3 that none of the knowledge items has a statistically significant association to “change in recommendation” may then serve as a warning that maybe the individual knowledge items are not very important factors to this outcome, that there are other factors that are better determinants for “change in recommendation” (which is not unreasonable to assert), or that not a single item, but rather several items determine the outcome. The logical premise of the relative importance approach is that IF we consider the effects of the items on an outcome important, WHAT is then the importance of these effects relative to each other (notably accounting for the intercorrelatedness of the items; something that is not taken into account in the marginal associations listed in Table 3). This will give an answer, even if none of the effects may be considered important in any absolute sense. All in all we think that the relative importance results give a good, and intuitive, indication of which items feed into the outcome most; the tables would be rather convoluted if we were to omit the relative importance for the non-significant results.”

---

## [Decision Letter · Decision Letter 3]

6 Sep 2024

PONE-D-22-34061R3Mammography screening: eliciting the voices of informed citizensPLOS ONE

Dear Dr. Jensen,

Thank you for submitting your manuscript to PLOS ONE. After careful consideration, we feel that it has merit but does not fully meet PLOS ONE’s publication criteria as it currently stands. Therefore, we invite you to submit a revised version of the manuscript that addresses the points raised during the review process.

We look forward to receiving your revised manuscript.

Kind regards,

Felix G. Rebitschek

Academic Editor

PLOS ONE

Journal Requirements:

Additional Editor Comments:

Dear Authors,

In response to a discussion with both the editorial board and a fourth reviewer on the dominance analyses performed in your study, finally, the following information should be added:

- provide a lot more details regarding the procedure employed for their DA, "in particular the regression models [..][that] preceded the DAs" and the parameters used to determine the ranking of coefficients.

- The reviewer notices "whether the authors are referring to general, conditional, or complete".

My general point is the following: Please consider the update in terms of a reproducibility perspective from potential readers.

Reviewers' comments:

Reviewer's Responses to Questions

**Comments to the Author**

1. If the authors have adequately addressed your comments raised in a previous round of review and you feel that this manuscript is now acceptable for publication, you may indicate that here to bypass the “Comments to the Author” section, enter your conflict of interest statement in the “Confidential to Editor” section, and submit your "Accept" recommendation.

Reviewer #4: (No Response)

2. Is the manuscript technically sound, and do the data support the conclusions?

Reviewer #4: (No Response)

3. Has the statistical analysis been performed appropriately and rigorously? 

Reviewer #4: (No Response)

4. Have the authors made all data underlying the findings in their manuscript fully available?

Reviewer #4: (No Response)

5. Is the manuscript presented in an intelligible fashion and written in standard English?

Reviewer #4: (No Response)

6. Review Comments to the Author

Reviewer #4: (No Response)

7. PLOS authors have the option to publish the peer review history of their article (what does this mean?). If published, this will include your full peer review and any attached files.

Reviewer #4: No

---

## [Author Response · Author response to Decision Letter 3]

28 Nov 2024

Journal Requirements:

Thank you for the comment. I have been through the reference list again and I am not aware that I have cited papers that have been retracted. Please let me know if you are aware of any retractions and I will correct it at once. 

Additional Editor Comments:

Dear Authors,

In response to a discussion with both the editorial board and a fourth reviewer on the dominance analyses performed in your study, finally, the following information should be added:

- provide a lot more details regarding the procedure employed for their DA, "in particular the regression models [..][that] preceded the DAs" and the parameters used to determine the ranking of coefficients.

- The reviewer notices "whether the authors are referring to general, conditional, or complete".

My general point is the following: Please consider the update in terms of a reproducibility perspective from potential readers.

Thank you for the comment. In the revised manuscript we describe the DA in more detail.

---

## [Editor Report · Decision Letter 4]

26 Dec 2024

Mammography screening: eliciting the voices of informed citizens

PONE-D-22-34061R4

Dear Dr. Jensen,

We’re pleased to inform you that your manuscript has been judged scientifically suitable for publication and will be formally accepted for publication once it meets all outstanding technical requirements.

Kind regards,

Felix G. Rebitschek

Academic Editor

PLOS ONE
---

## [Editor Report · Acceptance letter]

30 Dec 2024

PONE-D-22-34061R4 

PLOS ONE

Dear Dr. Jensen, 

I'm pleased to inform you that your manuscript has been deemed suitable for publication in PLOS ONE. Congratulations! Your manuscript is now being handed over to our production team.

Kind regards, 

on behalf of

Dr. Felix G. Rebitschek 

Academic Editor

PLOS ONE